# KCIF: Knowledge-Conditioned Instruction Following

**Rudra Murthy**
IBM Research AI
rmurthyv@in.ibm.com

**Praveen Venkateswaran**
IBM Research AI
praveen.venkateswaran@ibm.com

**Prince Kumar**
IBM Research AI
prince.kumar12@ibm.com

**Danish Contractor**
IBM Research AI
danish.contractor@ibm.com

## Abstract

LLM evaluation benchmarks have traditionally separated the testing of knowledge/reasoning capabilities from instruction following. In this work, we study the interaction between knowledge and instruction following, and observe that LLMs struggle to follow simple answer modifying instructions, and are also distracted by instructions that should have no bearing on the original knowledge task answer. We leverage existing multiple-choice answer based knowledge benchmarks and apply a set of simple instructions which include manipulating text (eg.: change case), numeric quantities (eg.: increase value, change formatting), operate on lists (eg.: sort answer candidates) and distractor instructions (eg.: change case of numeric answers). We evaluate models at varying parameter sizes (1B-405B) from different model families and find that, surprisingly, all models report a significant drop in performance on such simple task compositions. While large-sized and frontier models report performance drops of 40-50%, in small and medium sized models the drop is severe (sometimes exceeding 80%). Our results highlight a limitation in the traditional separation of knowledge/reasoning and instruction following, and suggest that joint-study of these capabilities are important. We release our benchmark dataset, evaluation framework code, and results for future work.

## 1 Introduction

The need for highly accurate and controllable LLM-powered systems that follow precise instructions have led to the development of datasets and methods to improve reliability and consistency in the output for LLMs. Such methods include few-shot prompting [Gao et al., 2020, Kojima et al., 2022b], reasoning with explanations [Wei et al., 2022, Huang and Chang, 2022], checking for consistency/self-consistency [Wang et al., 2022], use of intermediate evaluators or LLMs operating as judges [Zheng et al., 2023], and more. However, despite such rapid progress and 'benchmark saturation' LLMs can often struggle on very simple tasks. For instance, as shown in Figure 1, given a multiple-choice question with option labels and their text, if we instruct the models to print the text associated with the correct answer instead of the answer label, we observe a significant drop ($\sim 20\%$ on average) in knowledge-task performance.

From a user's perspective, this task is no harder than selecting (generating) the answer label. Yet, this pattern is consistent even for frontier models like *GPT-4o* (Appendix Figure 5) and such patterns of interaction where keys or values are referred to LLM instructions are fairly common in real-world LLM usage – for instance, in manipulating lists (eg: "replace the third bulleted item with 'x'" ), operating on semi-structured objects (eg: extracting the value of a JSON field and use in a subsequent operation), etc. While this example of a drop in performance when printing the answer-text instead of the answer-label could, be due to the training process resulting in models being over-fit to certain

Preprint. Under review.

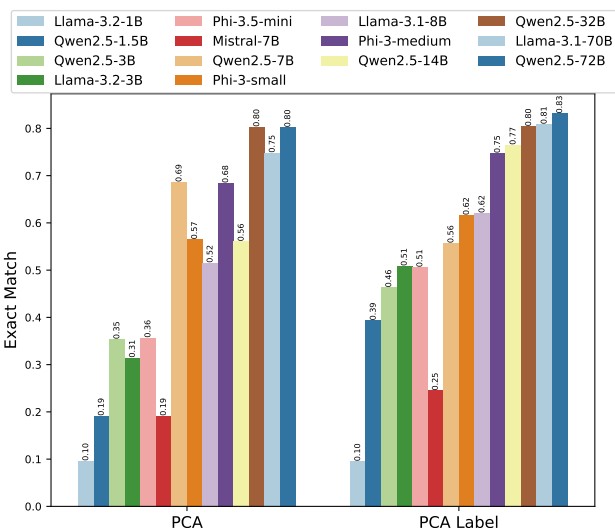

Figure 1: Average exact match performance across all tasks for the *print_correct_answer* (PCA) and *print_correct_answer_label* (PCA Label) instructions.

input/output task formats, or due to issues associated with label binding [Xue et al., 2024], there are other patterns of failure as well, – for instance, including additional *distractor* instructions that have no effect on the answering task, results in a further 5-30% drop (Section 4.2.3).

In this paper, we study this interaction of knowledge and instruction following; real-world use of LLMs typically involves both aspects, and '*nested*' instructions where there is conditional branching of instructions based on intermediate steps is a common usage pattern of LLMs. However, evaluating these reliably can be hard and can require the use of complex LLM-judges [He et al., 2024]. Thus, to keep evaluation easy and verifiable, our tasks are based on multiple-choice answers instead of long-form text generation task, We adapt commonly used knowledge and reasoning benchmarks including MMLUPro [Wang et al., 2024], MathQA Amini et al. [2019], Winogrande [Sakaguchi et al., 2021], BoolQ [Clark et al., 2019], PIQA Bisk et al. [2020] and augment them with two broad classes of instructions: (i) Instructions that are conditional on the answer to the question (ii) Instructions that are applied uniformly regardless of the answer or task. Our instructions are simple and include those that manipulate the text (eg.: change case), numeric quantities (eg: increase value, change formatting), operate on lists (eg: sort answer candidates). Our choice of using multiple-choice answers-based tasks with simple answer-modifying instructions, also allows us to *automate* error analysis for commonly occurring error-types and classify errors based on their origin - knowledge/reasoning errors vs instruction-following errors. Our results show that we can automatically classify approximately 80% of errors.

**Contributions:** In summary, we make the following contributions: (i) We present an evaluation framework that supports LLM-free scoring to study knowledge-conditioned instruction-following, (ii) Our novel experimental design enables automated error analysis (to the extent that LLMs dont have catastrophic generation failures), (iii) We create an evaluation benchmark using 5 popular datasets, consisting of 13 instruction types across 5 different categories. It can be easily extended to additional tasks and instructions, (iv) We use this benchmark to evaluate 19 models at varying parameter sizes (1B-405B) from different model families and find that, all models studied report a significant drop in performance on simple single-instruction task composition. While large-sized and frontier models report performance drops of 40-50%, in small and medium sized models the drop is severe (sometimes exceeding 80%). We also observe that, even within a model family, a larger parameter model may perform comparably to its smaller sibling on standard knowledge/reasoning tasks, but smaller ones drop more significantly on in our evaluation tasks. We release code[1] and data[2] to support evaluation, enable task extensions and future work.

---

[1] https://github.com/IBM/KCIF
[2] https://ibm.biz/KCIFData

## 2    Related Work

Evaluating the capabilities of large language models (LLMs) has been a significant area of research, with studies focusing on various aspects of LLM performance. Researchers have developed multiple benchmarks to assess factual knowledge Petroni et al. [2019], Roberts et al. [2020], Lin et al. [2022], logical reasoning abilities Wei et al. [2022], Zhou et al. [2023a], Saparov et al. [2023], general problem-solving capabilities Kojima et al. [2022a] and more.

Recently there have also been studies on instruction-following - for instance, FoFo Xia et al. [2024] evaluates models on format-following tasks and studies the ability of LLMs to generate outputs in existing real-world formats. In a similar vein, IFEval Zhou et al. [2023b] assesses LLMs' ability to follow arbitrary task-specific instructions (e.g.) based on response length, casing, etc, focusing primarily on whether the instructions are followed rather than the correctness of the output for the task. InFoBench Qin et al. [2024] advances this research by introducing a metric known as the 'Decomposed Requirements Following Ratio' (DRFR) which is based on each aspect of an instruction that needs to be met. Along with 500 diverse instructions and 2,250 decomposed questions, InFoBench offers performance evaluation using OpenAI's GPT4, across multiple constraint categories and highlights key areas where advanced LLMs can improve in complex instruction-following tasks. LLMBar [Zeng et al., 2024] is another contribution to this area, as it provides a meta-evaluation benchmark specifically designed to test an LLM evaluator's ability to discern instruction-following outputs. The benchmark consists of 419 manually curated pairs of outputs, where one output adheres to instructions and the other, while potentially more engaging or deceptive, does not. ComplexBench [Wen et al., 2024] and FollowBench [Jiang et al., 2024] aim to assess the ability of LLMs to follow arbitrary task specific instructions, while Meowpass [He et al., 2024] presents a benchmark to evaluate complex instruction following by dynamically composing multiple IFEval style instructions, and consists of 1500 diverse prompt and instruction pairs. Another study introduces ManyIFEval [Harada et al., 2025], a benchmark assessing LLMs' ability to follow multiple instructions, revealing a "curse of instructions" where accuracy declines as instruction count increases. Li et al. [2024] propose a method to evaluate instruction following ability via verbalizer manipulation. Specifically, they modify the classification task labels with different verbalizers which may or may not be semantically relevant to the task. They observe that all models fail to follow instructions when they instruct the model to flip the labels (unnatural setting).

Our work complements these efforts by developing a benchmark that allows for easy verification of both task performance and instruction-following capabilities simultaneously. We augment existing knowledge benchmarks by creating instructions that are *conditional* on answering the QA-based knowledge task correctly. We also include instructions that are applied on the candidate space of answers provided in these knowledge tasks. Our approach of applying instructions on knowledge tasks provides an easy way of measuring performance as well as automating error analysis. Further, it also allows us to study the interactions between knowledge and instruction following, and to investigate whether instructions serve as distractors for the original knowledge task when the instructions should result in no change to the original answer of the knowledge task.

## 3    Dataset Creation

### 3.1    Knowledge and Reasoning Tasks

Our framework supports including any knowledge task with a fixed answer space - we select the following commonly used benchmarks: MMLUPro [Wang et al., 2024], MathQA [Amini et al., 2019], BoolQ [Clark et al., 2019], PIQA [Bisk et al., 2020], and Winogrande [Sakaguchi et al., 2021] as the basis for our knowledge-grounded instruction-following benchmark. These datasets involve either binary classification or multiple-choice-questions (MCQs) spanning different reasoning and problem-solving skills. We select 1500 samples randomly from *each* dataset and apply answer-modifying instructions as described in the next section.

### 3.2    Instruction Categories

Unlike datasets that require open-ended generation for answering, our selected tasks have a structured answer-space. This allows us to craft instructions using these answer-spaces in a way that can be

| Instruction Group | Name | Definition | # Instances Full | Lite |
|---|---|---|---|---|
| String Manipulation | alternate_case_correct_answer | Print the text corresponding to the correction candidate answer of knowledge task in alternate case | 7867 | 950 |
| | capitalize_correct_answer | Print the text corresponding to the correct candidate answer of the knowledge task in upper case. | 7867 | 950 |
| | reverse_correct_answer_alternate_case | Reverse the text corresponding to the correct candidate answer of the knowledge task and print it in alternate case. | 9573 | 1383 |
| | reverse_correct_answer | Print the text corresponding to the correct answer in reverse | 7868 | 951 |
| Format Correct Answer | numformat_numeric_answer | Apply a specified decimal formatting the correct answer if it a is numeric quantity, otherwise print the correct answer as is. | 11336 | 1600 |
| | print_correct_answer_in_words | If the correct answer is a numeric quantity, display the numeric quantity in words, otherwise print the correct answer as is. | 9874 | 1320 |
| | print_correct_answer_append_string | Append a pre-specified string to the text associated with the correct candidate answer. | 7867 | 950 |
| Operations on List (Conditional on Correct Answer) | increment_incorrect_numeric_answers_by_one | If the candidate answer values are numeric quantities increment them by one and show them as a list. Other value types are not modified. | 7117 | 825 |
| | sort_only_incorrect_answers | Sort the candidate answers that are incorrect in ascending order | 7867 | 950 |
| | use_incorrect_options_to_create_string | Sort the incorrect candidates in ascending order and take the last character of the text associated with each incorrect option to create a string | 7868 | 951 |
| Operations on List (OOL) | sort_options_to_create_string | Sort all candidate answers in ascending order and use the last character of the text associated with each incorrect candidate to create a string. | 7867 | 950 |
| Numeric Manipulation | increment_correct_numeric_answer_by_one | If the correct answer is a numeric quantity, increment it by one, otherwise print the correct answer as is. | 9757 | 1352 |

Table 1: Categories of instructions and the number of instances of each in the Full and Lite subsets.

verified easily. We summarize our instruction categories in Table 1. The task prompts (instructions) for each of the 13 instruction types with an example are available in the Appendix (Section B.4).

**Instruction Creation:** To create each instruction, the authors iteratively refined them until all the authors had complete agreement in the output when they followed them manually. Examples of aspects of iterative improvement include - explicitly making clear what is not to be included in the output, how the output is to be presented, etc. We include additional details of the instruction-writing process in the Appendix (B.2).

**Answering baseline-instructions:** Given the multiple-choice answer knowledge benchmarks, we consider two baseline instructions – (1) printing the correct answer option[3] from the candidate space ($print\_correct\_answer\_label$), and (2) printing only the text associated with the correct answer option ($print\_correct\_answer\_text$).

**Instructions with no-effect**: Certain instructions may be inapplicable for some knowledge tasks. For example, in the MathQA dataset, some instances have *none of these* as the correct answer and are not numeric. Here, instructions such as $numformat\_numeric\_answer$ or $increment\_correct\_numeric\_answer\_by\_one$ will not affect the existing answer of the knowledge-task. We refer to these instructions as "*distractor*" instances and expect that in these instances, models should perform as well as they do on the original answering task. We include details and statistics of such instructions in the Appendix (Section B.7 Tables 7 and 9).

## 3.3 Metrics and Automated Error Classification

We report the model performance as a form of exact match (EM) where we perform basic string parsing (removing beginning and ending whitespaces, quotations, etc.) and compare the model prediction to the expected output for the applied instruction. See Appendix Section B.5 for more details.

For each instruction in the 'String Manipulation', 'Format Correct Answer', and 'Numeric Manipulation' instruction categories, we create a set of error classes that are based on the incorrect answers to the original knowledge task and the subsequent application of the instruction on that incorrect answer. We create 'instruction-following' error sets (IFError) and 'knowledge-error sets' (KnowledgeError) as follows:

**IFError**: We inspected some representative model outputs and created a set of errors that LLMs could plausibly make for each instruction. This set includes errors such as: (i) answering with an answer label (option) when requiring/using the text, (ii) answering with any candidate answer text without applying the instruction, (iii) a combination of these, (iv) instruction specific errors based

---

[3]We use 'label' and 'option' interchangeably to denote the candidates in a multiple-choice QA task.

| Small ($< 7B$ parameters) | Medium ($7 - 30B$ parameters) | Large ($> 30B$ parameters) | Frontier |
|---|---|---|---|
| Llama-3.2-1B-Instruct (1B) | Mistral-7B-Instruct-v0.3 (7B) | Qwen2.5-32B-Instruct (32B) | Llama-3.1-405B-Instruct (405B) |
| Qwen2.5-1.5B-Instruct (1.5B) | Qwen2.5-7B-Instruct (7B) | Llama-3.1-70B-Instruct (70B) | GPT-4o-mini-2024-07-18 |
| Llama-3.2-3B-Instruct (3B) | Phi-3-small-8k-instruct (7B) | Qwen2.5-72B-Instruct (72B) | GPT-4o-2024-08-06 |
| Qwen2.5-3B-Instruct (3.0B) | Llama-3.1-8B-Instruct (8B) | | |
| Phi-3.5-mini-instruct - (3.8B) | Gemma-2-9b-it (9B) | | |
| | Phi-3-medium-4k-instruct (14B) | | |
| | Qwen2.5-14B-Instruct (14B) | | |
| | Gemma-2-27b-it (27B) | | |

Table 2: List of Models evaluated on our benchmark.

on these – for example, for the instruction $alternate\_case\_correct\_answer$ such error cases could include alternate casing starting with a lowercase character (the instruction text explicitly requires that it start with upper case).

**KnowledgeError:** For every instruction instance, this set includes: (i) Answering with the incorrect candidate answer – regardless of whether it is its label or text, (ii) Applying the instruction correctly (when applicable) on the incorrect candidate answer.

Note that these error sets are not mutually exclusive – a model response to the alternate casing instruction that starts with lowercase and is applied on the incorrect answer candidate is an 'IFError' as well as a 'KnowledgeError'. Further, these error sets are meant to be *high-precision* to aid analysis and cannot reasonably expected to anticipate and cover everything LLMs might respond with. Hence, there will be model errors that we cannot classify[4] and we intend to continue to expand the benchmark's error detection capabilities. We would also like to highlight that if models make a lot of errors that are not classified, it is likely that those are more severe instruction-following errors.

## 3.4 Benchmark Dataset

We create two versions of our benchmark dataset - 'Full' and 'Lite' (for lower inference costs).

**Full Benchmark**: We select a subset of 1500 samples randomly from *each* dataset and apply each applicable instruction on the same. For MMLUPro, we consider a subset of 150 samples per subject and apply each applicable instruction.

**Lite Benchmark:** We select a subset of 150 samples randomly from the full version created above for *each* dataset and apply each applicable instruction on the same. For MMLUPro, we consider a subset of 25 samples per subject and apply each applicable instruction. Statistics for the above two versions are available in presented in Table 1. Detailed statistics for each dataset and the instruction types are provided in the appendix section B.7. Additionally, each benchmark includes a set of instances when instructions have no effect (called the no-effect or distractor subset).

### 3.4.1 Benchmark ranking

An effective instruction-following model should not only be capable of following a variety of instructions across different knowledge-tasks but should also be unaffected by instructions when they are inapplicable i.e, they should be robust to 'distractors'. Therefore, we define an overall benchmark score for a model as its arithmetic mean of the following:

**Exact-Match Score ($\mu_{EM}$):** We compute the micro-average of the exact-match scores using all instances of every instruction type in the benchmark.

**Instruction Category Score (IC Score)**: To ensure models perform well across instruction categories, we also compute the micro-average exact-match scores for every instance per instruction category and then compute the arithmetic mean.

**Knowledge Task Subset Score: (KTS Score):** To ensure model capabilities generalize across different knowledge tasks, we compute the micro-average exact-match scores for every instance per knowledge-task, and then compute the arithmetic mean.

---

[4]Errors are matched using the same output-processing used for ground-truth scoring.

**Exact Match Score on 'Instructions with no-effect'** ($\mu'_{EM}$): Lastly, we compute the micro-average of all instruction instances in the benchmark that have no effect on the original knowledge-task answers (i.e.) 'distractors'.

## 4 Evaluation

We present an evaluation on our benchmark using a variety of models and study the following research questions: (i) Do models display a difference in performance on the two simple answering baseline-instruction tasks? (ii) Do models display a variation in performance across our different instruction categories? (iii) Are models robust to, or get distracted by instructions that do not apply to the task? (iv) Does the size of a model impact its instruction-following capability?

### 4.1 Models and Inference

We evaluate a range of open instruction-tuned models and parameter sizes using our benchmark. For ease of presentation, we categorize them based on their parameter count as shown in Table 2. Our inference code uses vLLM Kwon et al. [2023] for running the evaluations. We use greedy decoding for generations and `bf16` as floating point precision. We generate a maximum of 1024 tokens per instance. We use A100 80GB GPUs for running inference. We use an instance hosted by a cloud provider for Llama-3.1-405B-Instruct, while we use OpenAI APIs for GPT4-o and GPT4-o-mini models.

In all our experiments, we perform zero-shot Chain-of-Thought (CoT) Wei et al. [2024] reasoning. Models see the same prompt based on prompt guides for the original knowledge tasks using the *lm-evaluation-harness* framework Gao et al. [2024] and OpenAI evals.[5] We instruct the model to generate reasoning first and then the answer (See examples in Appendix Section B.4). We write custom post-processing scripts to extract the model's answer as described previously.

| Model | PCA label | PCA | String | Numeric | Format | OOL Conditional | OOL | Avg. Drop |
|---|---|---|---|---|---|---|---|---|
| Qwen2.5-1.5B-Instruct | 0.36 | -0.20 | -0.35 | -0.34 | -0.32 | -0.29 | -0.35 | -0.33 / -90.5% |
| Qwen2.5-3B-Instruct | 0.44 | -0.12 | -0.39 | -0.42 | -0.2 | -0.3 | -0.24 | -0.31 / -69.7% |
| Qwen2.5-7B-Instruct | 0.55 | 0.11 | -0.41 | -0.11 | -0.33 | -0.32 | -0.38 | -0.31 / -56.5% |
| Qwen2.5-14B-Instruct | 0.75 | -0.20 | -0.60 | -0.21 | -0.57 | -0.30 | -0.29 | -0.39 / -52.8% |
| Qwen2.5-32B-Instruct | 0.79 | -0.01 | -0.47 | -0.17 | -0.40 | -0.27 | -0.26 | -0.31 / -39.9% |
| Qwen2.5-72B-Instruct | 0.81 | -0.04 | -0.50 | -0.25 | -0.31 | -0.27 | -0.18 | -0.30 / -37.3% |
| Llama-3.2-1B-Instruct | 0.10 | -0.01 | -0.08 | -0.08 | -0.09 | -0.07 | -0.10 | -0.08 / -87.6% |
| Llama-3.2-3B-Instruct | 0.47 | -0.18 | -0.37 | -0.37 | -0.36 | -0.37 | -0.45 | -0.38 / -81.5% |
| Llama-3.1-8B-Instruct | 0.59 | -0.09 | -0.47 | -0.31 | -0.37 | -0.42 | -0.45 | -0.40 / -67.7% |
| Llama-3.1-70B-Instruct | 0.79 | -0.06 | -0.42 | -0.39 | -0.41 | -0.41 | -0.43 | -0.41 / -52.5% |
| Phi-3.5-mini-instruct | 0.48 | -0.14 | -0.43 | -0.33 | -0.35 | -0.37 | -0.34 | -0.36 / -75.5% |
| Phi-3-small-8k-instruct | 0.58 | -0.05 | -0.47 | -0.43 | -0.34 | -0.42 | -0.52 | -0.44 / -74.8% |
| Phi-3-medium-4k-instruct | 0.72 | -0.07 | -0.53 | -0.50 | -0.46 | -0.37 | -0.40 | -0.45 / -63.2% |
| Gemma-2-9b-it | 0.62 | -0.05 | -0.50 | -0.37 | -0.29 | -0.30 | -0.30 | -0.35 / -56.5% |
| Gemma-2-27b-it | 0.69 | -0.08 | -0.44 | -0.37 | -0.30 | -0.23 | -0.22 | -0.31 / -45.3% |
| GPT4-o-mini | 0.74 | -0.14 | -0.39 | -0.36 | -0.35 | -0.31 | -0.30 | -0.34 / -45.8% |
| GPT4-o | 0.81 | -0.04 | -0.32 | -0.32 | -0.37 | -0.28 | -0.18 | -0.29 / -36.2% |
| Llama-3.1-405B-Instruct | 0.83 | -0.07 | -0.39 | -0.41 | -0.41 | -0.39 | -0.45 | -0.34 / -41.0% |

Table 3: As compared to printing the correct answer label (PCA Label) which serves as the baseline for the vanilla knowledge task, printing the correct answer (PCA) results in a significant drop in exact match performance. Applying additional instructions on the answer space of the questions result in an even more severe drop (on every category). Drop in performance reported as compared to PCA Label. Frontier model results reported on Lite Benchmark data.

### 4.2 Results

We begin this section by first presenting our results on the answering baseline-instructions and then proceed to our results on instruction-following for the different categories. We then look at the impact of distractors and knowledge-task characteristics on model performance.

---

[5] `https://github.com/openai/simple-evals`

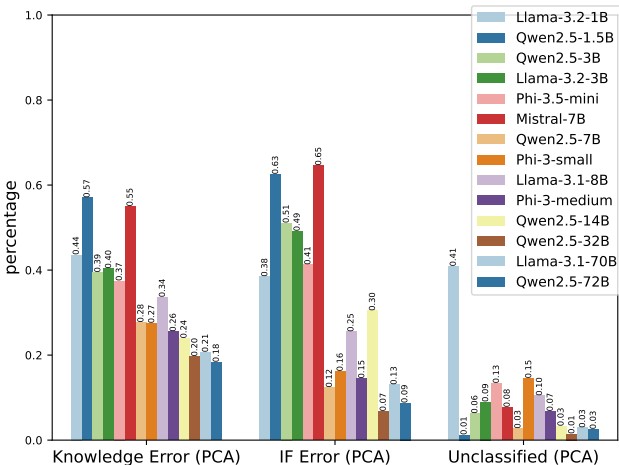

Figure 2: Knowledge and instruction following (IF) errors across all tasks for the *print_correct_answer* instruction. A lower error is better. Results shown using Full Benchmark data. Lite Benchmark results can be found in Appendix Figure 5.

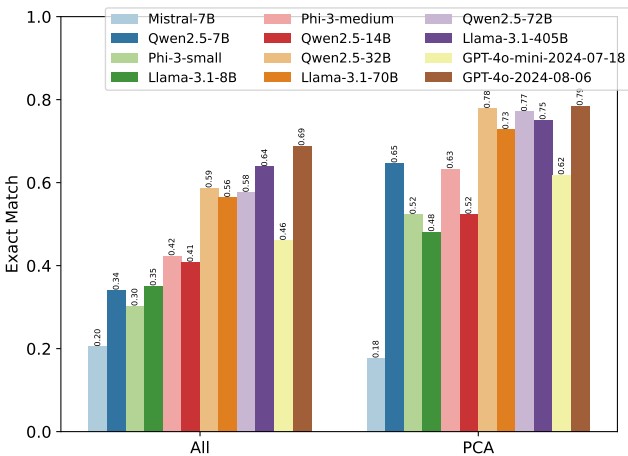

Figure 3: Impact of distractor instructions on exact match performance across tasks and instructions, compared to its corresponding $print\_correct\_answer$ performance. A drop indicates the model getting distracted by an inapplicable instruction. Results reported on Lite Benchmark.

### 4.2.1 Printing the correct answer

As mentioned earlier in Section 1 (Figure 1), we observe a significant drop ($\sim 20\%$ on average) in knowledge-task performance when instructing the model to respond with the text associated with the answer instead of its label. Some common issues we observed include models outright ignoring the instruction and continuing to generate labels, or generating only Chain-of-Thought reasoning without a final answer, etc, reflected by the knowledge and instruction following errors in Figure 2.

### 4.2.2 Answer-conditioned instruction performance

Table 3 reports the performance of all models on the print correct answer label (PCA Label) task which serves as the reference score for the knowledge/reasoning task.

**Effect of Size Models:** Starting with the Qwen 2.5 family of models it can be seen that in medium and large sized models (>14B parameter models) have comparable performance on the baseline answering task (PCA label). The drop for the large models (> 32B parameters) on the PCA task is relatively small as well. However, the drop in scores when combined with instructions result is much larger, and a trend is observed that larger models still do better than their smaller siblings. This holds

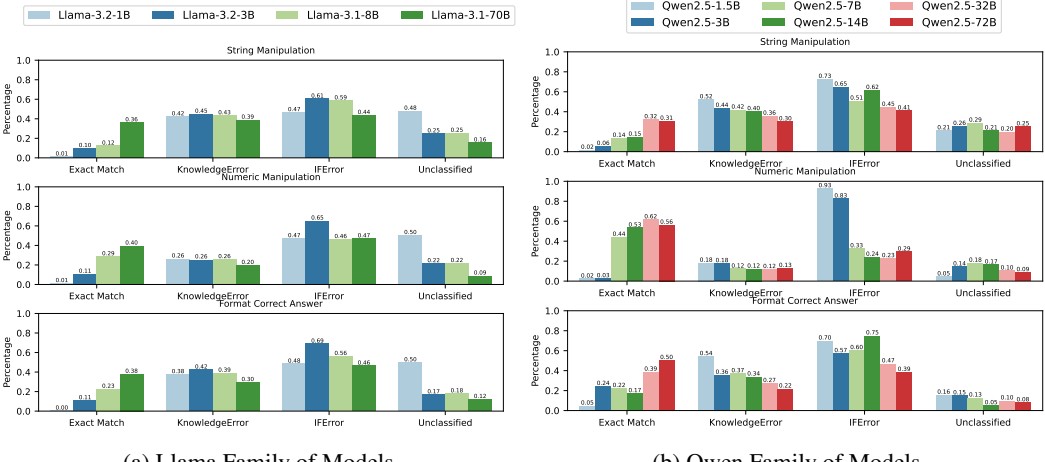

(a) Llama Family of Models        (b) Qwen Family of Models

Figure 4: Classification of errors for the Llama and Qwen family of models.

true in all model families (Llama, Gemma and Phi) - even if there is a negligible difference in the PCA label performance of a model, the drop is less severe for a larger sibling from the same family. This result demonstrates that even if models perform comparably on the PCA label task, the large model size can benefit tasks which require knowledge and instruction composition.

**Performance of Frontier Models:** From Table 3 we find that frontier models also suffer from significant drops in performance. Notably, GPT4-o and Llama 3.1-405B Instruct have comparable performance on the knowledge task (as seen on PCA Label) but Llama's performance deteriorates by a larger degree when combined with instructions.

### 4.2.3 Effect of distractor instructions

Our dataset also includes instructions that apply only when certain properties of a knowledge-task answer are fulfilled. For instance, instructions for incrementing the correct answer by one if numeric, formatting numeric values, and printing any numeric answers in words, do not apply on tasks with textual answers. They serve as distractors, and we expect model performance to be unaffected since these instructions are not applicable and do not alter the original knowledge-task answer. From Figure 3, we observe that there is a 5-20% drop in small, medium, large, and frontier scale models.

In the appendix Figures 6, 7, 8, we report details of how different model families (Llama, Qwen, Phi) are affected by distractors, at different scales. In summary, we find that all models are distracted by instructions that require reversing and casing text,[6] reporting a drop of 55-75%. Even frontier models (Appendix Figure 9) show a drop of 35%. However, for the other distractor instructions, we find that Llama models are relatively robust, showing a drop of $5-10\%$, while the Phi and Qwen family of models are more susceptible, where we observe drops of 20-30%. Frontier models surprisingly also show a drop of 10-15%. While model failures in the presence of distractors have been studied before [Shi et al., 2023, Feng et al., 2024], to the best of our knowledge this is the first work to study them in an instruction-*following* setting.

### 4.2.4 Automated Error Classification

Figure 4 shows the error analysis for two different model families - Llama and Qwen2.5.[7] We report the average exact match score as well as the errors.

**Errors and Model size:** As can be seen in both Llama and Qwen2.5 model families, as the model size increases, the exact match accuracy increases and the errors reduce. The unclassified errors pertain to model outputs that are harder to classify and indicate a more *severe* model failure. As, the scale increases, such failures tend to reduce.

---

[6]partially applicable (can only reverse non-single digit numeric answers) or completely inapplicable

[7]We include error analysis plots for other model families in the appendix.

| Models | $\mu_{EM}$ | IC Score | KTS Score | $\mu'_{EM}$ | Average Score |
|---|---|---|---|---|---|
| GPT-4o-2024-08-06 | 0.5065 | 0.5174 | 0.5874 | 0.6889 | 0.575 |
| Llama-3.1-405B | 0.4617 | 0.4888 | 0.5351 | 0.6387 | 0.5311 |
| Qwen2.5-72B | 0.4348 | 0.5035 | 0.493 | 0.5768 | 0.502 |
| Qwen2.5-32B | 0.409 | 0.4751 | 0.4755 | 0.5873 | 0.4867 |
| Llama-3.1-70B | 0.3708 | 0.4138 | 0.4319 | 0.5645 | 0.4453 |
| GPT-4o-mini | 0.394 | 0.4029 | 0.4689 | 0.4609 | 0.4317 |
| Gemma-2-27B | 0.3497 | 0.3972 | 0.4194 | 0.4505 | 0.4042 |
| Qwen2.5-14B | 0.2764 | 0.3523 | 0.3272 | 0.4084 | 0.3411 |
| Phi-3-medium | 0.2518 | 0.2869 | 0.3054 | 0.4238 | 0.317 |
| Gemma-2-9B | 0.2381 | 0.2828 | 0.292 | 0.4428 | 0.3139 |
| Qwen2.5-7B | 0.1944 | 0.2513 | 0.2275 | 0.3411 | 0.2536 |
| Llama-3.1-8B | 0.174 | 0.2203 | 0.2048 | 0.3513 | 0.2376 |
| Phi-3-small | 0.1555 | 0.1809 | 0.1921 | 0.3027 | 0.2078 |
| Mistral-7B | 0.0577 | 0.0808 | 0.0768 | 0.205 | 0.1051 |

Table 4: Performance of the Medium, Large and Frontier Models on our Lite Benchmark - ranked in order of performance using the average score (higher is better).

**Instruction Categories and Errors:** We note from Figure 4, the Llama and Qwen2.5 models have different strengths. For instance, in String Manipulation class of instructions, at very small scale the Llama 3.2-1B model and the Qwen2.5-1B model have similar exact match scores. However, in the case of the Llama model nearly 48% of all instances have errors that are unclassified (indicating severe failure) while that is not the case for the Qwen2.5 equivalent. Further, Qwen2.5-1B has a larger class of instruction-following errors as compared to knowledge errors; this suggests that the drop in performance for Qwen2.5-1B reported in Table 3 is indeed due to instruction-following errors made by the model.

In addition, we observe that the Qwen2.5 family of models in almost always better than the Llama family of models when it comes to numeric manipulation instructions while the Llama family of models has a slightly higher performance on string manipulation tasks (Appendix Figures 39 and 40). A similar trend is observed on the unclassified set where Llama's instruction-following failures appear to be more severe. The Phi family of models (Appendix Figure 34) appears to perform worse than both the Qwen and Llama family of models. Frontier models (Appendix Figure 38) demonstrate lower knowledge errors as expected, but still make many instruction following errors. We include a qualitative example of each error category in Appendix Section B.6.

### 4.3 Benchmark

We report the *exact match* scores of the medium, large and frontier models on the Lite Benchmark in Table 4. Unsurprisingly, the GPT4o model performs the best on our benchmark data while large and medium-scale models like Llama-3.1 405B, Qwen2.5 72B, and, Qwen2.5 32B based models appear to be better than other openly available models including Llama-3.1-70B-instruct and the Gemma family of models. We also include the results on the full benchmark in Appendix Table 5. We note that the ranking of models is largely consistent and that small models are much weaker than larger models.

## 5 Discussion & Conclusion

In this work, we took a novel approach to studying instruction-following by grounding instructions on existing knowledge tasks. Our approach has the advantage of being easily extendable for new instruction types and domains, while also enabling LLM-free evaluations with some degree of automated error analysis. We demonstrated that not only do models fail to follow simple instructions (e.g.) printing the answer text instead of the label, but their performance drops further when compound but simple, instructions are included. Even when instructions that should have no effect on the knowledge-tasks are used, models at all scales report a drop in performance, though the extent of deterioration varies. As models are increasingly being viewed as agents and assistants, it is crucial that models have better guarantees of following user instructions. As our work demonstrates, there is a lot of scope for improvement and we hope the community finds our framework helpful in improving the current state-of-the-art.

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

# A    Appendix

We begin by including details of the knowledge tasks used. We then describe our data creation process and how we automatically classify errors in section 3.3. We list all instructions with example input, ground truth, and expected instruction output in B.4. We report results on the Full Benchmark in Table 5. The detailed statistics of Full and Lite Benchmark are presented in B.7. Section D.1 presents the comparison between model's performance on print correct answer and print correct answer labels tasks on the Lite Benchmark. Section D.3 presents performance of different models for each instruction category in comparison with its corresponding performance on $print\_correct\_answer$ (PCA).

# B    Additional details - Dataset creation

## B.1    Design Principles

We develop our instructions keeping the following design principles in mind: (i) We would like instructions to be unambiguous and be presented in a way that can be communicated clearly - if

| Models | $\mu_{EM}$ | IC Score | KTS Score | $\mu'_{EM}$ | Average Score |
|---|---|---|---|---|---|
| Qwen2.5-72B | 0.4488 | 0.5077 | 0.4708 | 0.6218 | 0.5123 |
| Qwen2.5-32B | 0.419 | 0.4736 | 0.4519 | 0.6351 | 0.4949 |
| Llama-3.1-70B | 0.3697 | 0.3735 | 0.3925 | 0.6109 | 0.4366 |
| Gemma-2-27B | 0.3622 | 0.3783 | 0.3984 | 0.5177 | 0.4142 |
| Qwen2.5-14B | 0.2819 | 0.3521 | 0.305 | 0.4443 | 0.3458 |
| Phi-3-medium | 0.2589 | 0.2632 | 0.2799 | 0.4897 | 0.3229 |
| Gemma-2-9B | 0.2417 | 0.2701 | 0.2688 | 0.484 | 0.3162 |
| Qwen2.5-7B | 0.1921 | 0.2393 | 0.2 | 0.4061 | 0.2594 |
| Llama-3.1-8B | 0.1646 | 0.1917 | 0.1773 | 0.3907 | 0.2311 |
| Phi-3-small | 0.1472 | 0.1474 | 0.1686 | 0.3376 | 0.2002 |
| Qwen2.5-3B | 0.1277 | 0.1341 | 0.1386 | 0.3021 | 0.1756 |
| Llama-3.2-3B | 0.0946 | 0.0874 | 0.1021 | 0.2395 | 0.1309 |
| Phi-3.5-mini | 0.0966 | 0.1179 | 0.1014 | 0.2044 | 0.1301 |
| Mistral-7B | 0.0484 | 0.059 | 0.057 | 0.2451 | 0.1024 |
| Qwen2.5-1.5B | 0.0382 | 0.0346 | 0.0435 | 0.1461 | 0.0656 |
| Llama-3.2-1B | 0.0153 | 0.012 | 0.0176 | 0.0897 | 0.0337 |

Table 5: Performance of the Small, Medium, and Large Models on our Full Benchmark - models ranked in order of performance using the average score (higher is better).

humans cannot follow the instructions and agree on the same output, LLMs should and likely would not be able to. (ii) We would like them to be easy to follow and not require complex reasoning abilities to follow so that models at all scales have a fair chance of success, (iii) The instructions need to have deterministic outputs that use the original answers of the knowledge-task or the candidate space of answers, or both, so that they can be evaluated easily with instruction specific scorers. (iv) We would like our benchmark to be based on a diverse mix of knowledge tasks, and be easily extensible to new ones.

## B.2 Instruction Creation

To create each instruction, the authors iteratively refined them until all the authors had complete agreement in the output when they followed them manually. Examples of aspects of iterative improvement include - explicitly making clear what is not to be included in the output, how the output is to be presented, etc. We then asked 2 computer science researchers to follow and generate the output for 75 instructions across all our instruction types and datasets. We found that both the researchers were able to follow our instructions successfully and generated the same response for 93.33% of the instances. The first annotator generated the correct response for 98.67% of the instances, while the second annotator for 94.67% of the instances. Upon analyzing their responses, we found the only instruction-following error was rounding off the decimal number when truncating to two decimal places. We also found very few human errors in the annotator's response, specifically for instructions like *reverse_correct_answer_alternate_case* on datasets with long output text such as PIQA.

## B.3 Knowledge and Reasoning Tasks

We select the following knowledge tasks that are commonly used in LLM evaluations as the basis for our instruction-following benchmark. These datasets involve either binary classification or multiple-choice-questions (MCQs) spanning different reasoning and problem-solving skills.

(i) **MMLUPro** [Wang et al., 2024]: MMLUPro extends the MMLU dataset to make it more challenging by a) increasing the number of options from four to ten and b) increasing problem difficulty by focusing on more reasoning oriented problems. We consider all 14 subjects in the MMLUPro benchmark. We cap the maximum number of samples for each subject to be 150 samples.

(ii) **MathQA** [Amini et al., 2019]: MathQA dataset consists of math word problems presented as Multiple-Choice-Questions (MCQs). Given a math question and four options, the model has to select the correct answer.

We also select a few common-sense and reasoning datasets:

(iii) **BoolQ** [Clark et al., 2019]: BoolQ is a boolean question-answering dataset. Given a passage and a boolean question around the passage, the model has to select either *True* or *False*.

(iv) **PIQA** [Bisk et al., 2020]: Physical Interaction: Question Answering (PIQA) involves answering questions that involve commonsense reasoning around physical objects. Given a question and two options, the model has to select the most plausible option.

(v) **Winogrande** [Sakaguchi et al., 2021]: Winogrande involves a fill-in-the-blank task with binary options, the model has to select the correct option for a given sentence. The task involves reasoning for pronoun resolution.

We select a subset of 1500 samples randomly from each of the above datasets.

## B.4   Prompts

We now list the prompts for each instruction and dataset combination defined in our benchmark.

---

Dataset: Winogrande
Instruction: print_correct_answer_append_string

Given a sentence with a blank and 2 options namely 'A' and 'B', print the text associated with the option label that answers the question correctly. Do not print the option label. Without including any additional characters, also append the following text string at the end of the output:'ENDOFRESPONSE'. Do not add any special characters including quotations, spaces, new lines etc in the response. Think step by step and in the end, finish your response with 'Response:$RESPONSE' where $RESPONSE (without quotes) is the final output expected.
Sentence: It is an article of faith that the paper is more important than the exam , even though the _ weighs less heavily on the grade.
**Options:**
A. paper
B. exam
**Ground Truth:** A
**Instruction Output:** Response:paperENDOFRESPONSE

---

Dataset: Winogrande
Instruction: alternate_case_correct_answer

Given a sentence with a blank and 2 options namely 'A' and 'B', answer the question by printing the text associated with the correct option label in alternate case. The first character should be in uppercase and the following characters should alternate between lowercase and uppercase. Do not print the option label. Think step by step and in the end, finish your response with 'Response:$RESPONSE' where $RESPONSE (without quotes) is the final output expected.
Sentence: The wooden doors at my friends work are worse than the wooden desks at my work, because the _ material is stronger.
**Options:**
A. doors
B. desks
**Ground Truth:** B
**Instruction Output:** Response:DeSkS

---

Dataset: MathQA
Instruction: numformat_numeric_answer

Given a mathematical question and 5 options namely 'a', 'b', 'c', 'd', and, 'e', as candidate answers, print the text associated with the option label that answers the question correctly. If the answer is numeric print it in two decimal places as long as it contains no other string or units of measurement. Do not print the option label. Think step by step and in the end, finish your response with 'Response:$RESPONSE' where $RESPONSE (without quotes) is the final output expected.
**Question:** a man walking at the rate of 5 km / hr crosses a bridge in 15 minutes . the length of the bridge ( in meters ) is :
**Options:**
a. 600
b. 750
c. 1000
d. 1250
e. none of these
**Ground Truth:** d
**Instruction Output:** Response:1250.00

---

Dataset: MathQA
Instruction: sort_options_to_create_string

Given a mathematical question and 5 options namely 'a', 'b', 'c', 'd', and, 'e', as candidate answers, sort the list of options using their values, in alphabetical order. Use only the text associated with the option labels and not the option labels while sorting. Then, create a string by concatenating the last character of the text associated with each option value. If the last character is a special character (such as period, comma, quotation, etc) use the previous character. Print only the final string and not the sorted list. Think step by step and in the end, finish your response with 'Response:$RESPONSE' where $RESPONSE (without quotes) is the final output expected.
**Question:** marts income is 50 percent more than tims income and tims income is 40 percent less than juans income . what percentage of juans income is marts income
**Options:**
a. 124 %
b. 120 %
c. 96 %
d. 90 %
e. 64 %
**Ground Truth:** d
**Instruction Output:** Response:40604

---

Dataset: PIQA
Instruction: reverse_correct_answer

Given a question and two answer candidates 'A' and 'B', answer the question by printing the text associated with the correct option label, in reverse. Do not print the option label. Think step by step and in the end, finish your response with 'Response:$RESPONSE' where $RESPONSE (without quotes) is the final output expected.
**Question:** Butcher Shop
**Options:**
A. will decimate fish from the ocean into digestible pieces
B. will decimate a full cow into digestible pieces
**Ground Truth:** B
**Instruction Output:** Response:seceip elbitsegid otni woc lluf a etamiced lliw

Dataset: PIQA
Instruction: print_correct_answer

Given a question and two answer candidates 'A' and 'B', answer the question by selecting the value associated with the option label corresponding to the correct answer. Do not print the option label. Think step by step and in the end, finish your response with 'Response:$RESPONSE' where $RESPONSE (without quotes) is the final output expected.
**Question:** how to avoid paint spill when adding paint to your brush
**Options:**
A. Put a rubber band on your paint can to get rid of that excess glue on your paint brush, this will prevent spilling paint on the paint stir stick where the lid is.
B. Put a rubber band on your paint can to get rid of that excess glue on your paint brush, this will prevent spilling paint on the edge where the lid is.
**Ground Truth:** B
**Instruction Output:** Response:Put a rubber band on your paint can to get rid of that excess glue on your paint brush, this will prevent spilling paint on the edge where the lid is.

Dataset: Winogrande
Instruction: use_incorrect_options_to_create_string

Given a sentence with a blank and 2 options namely 'A' and 'B', create a string by concatenating the last character of every option value, excluding the option value corresponding to the correct answer. Use only the text associated with the option labels and not the option labels while sorting. If the last character is a special character (such as period, comma, quotation, etc) use the previous character. Think step by step and in the end, finish your response with 'Response:$RESPONSE' where $RESPONSE (without quotes) is the final output expected.
Sentence: Fiction books were interesting and easy to read for Logan but not Brett because _ enjoyed real tales.
**Options:**
A. Logan
B. Brett
**Ground Truth:** B
**Instruction Output:** Response:n

Dataset: PIQA
Instruction: reverse_correct_answer_alternate_case

Given a question and two answer candidates 'A' and 'B', reverse the text associated with the answer label that correctly answers the question. Print this reversed text in alternate case starting with upper case. Do not print the option label. Think step by step and in the end, finish your response with 'Response:$RESPONSE' where $RESPONSE (without quotes) is the final output expected.
**Question:** What else should I add to a peanut butter sandwich?
**Options:**
A. Take some pasta sauce and put it onto the other piece of bread with a knife.
B. Take some jelly and put it onto the other piece of bread with a knife.
**Ground Truth:** B
**Instruction Output:** Response:.EfInK A HtIw dAeRb fO EcEiP ReHtO EhT OtNo tI TuP DnA YlLeJ EmOs eKaT

Given a sentence with a blank and 2 options namely 'A' and 'B', create a string by concatenating the last character of every option value (not option label). If the last character is a special character (such as period, comma, quotation, etc) use the previous character. Think step by step and in the end, finish your response with 'Response:$RESPONSE' where $RESPONSE (without quotes) is the final output expected.
Sentence: Megan focused less on proper posture than Lindsey because _ wanted to become a model.
**Options:**
A. Megan
B. Lindsey
**Ground Truth:** B
**Instruction Output:** Response:ny

Given a mathematical question and 5 options namely 'a', 'b', 'c', 'd', and, 'e', as candidate answers, answer the question by selecting the option label corresponding to the correct answer. Do not include the text associated with the option label in the answer. Think step by step and in the end, finish your response with 'Response:$RESPONSE' where $RESPONSE (without quotes) is the final output expected.
**Question:** a reduction of 20 % in the price of salt enables a lady to obtain 2 kgs more for rs . 100 , find the original price per kg ?
**Options:**
a. 12.6
b. 12.1
c. 12.5
d. 12.4
e. 12.7
**Ground Truth:** c
**Instruction Output:** Response:c

Given a question and two answer candidates 'A' and 'B', print the text associated with the option label that answers the question correctly. Note that if the correct answer is a numeric quanity, including dollar values and percentages but contains no other string or units of measurement, print the value after increasing its value by 1. Dollar values should be prefixed with '$'. Do not print the option label. Think step by step and in the end, finish your response with 'Response:$RESPONSE' where $RESPONSE (without quotes) is the final output expected.
**Question:** how to winterize windows
**Options:**
A. put weather stripping around them to stop air from escaping and air from coming in
B. put weather stripping around them to stop air from escaping and air from coming into the dishwasher
**Ground Truth:** A
**Instruction Output:** Response:put weather stripping around them to stop air from escaping and air from coming in

Given a mathematical question and 5 options namely 'a', 'b', 'c', 'd', and, 'e', as candidate answers, excluding the option that answers the question correctly, print a sorted list (ascending order) of the incorrect options. Do not print the option labels. Use the text associated with the option labels and not the option labels while sorting and printing. Think step by step and in the end, finish your response with 'Response:$RESPONSE' where $RESPONSE (without quotes) is the final output expected.

**Question:** the sector of a circle has radius of 21 cm and central angle 108 o . find its perimeter ?

**Options:**
a. 81.6 cm
b. 85.9 cm
c. 90 cm
d. 92 cm
e. 95 cm

**Ground Truth:** a

**Instruction Output:** Response:['85.9 cm', '90 cm', '92 cm', '95 cm']

Given a question and two answer candidates 'A' and 'B', print the text associated with the option label that answers the question correctly. However, if the correct answer is a numeric value with no additional text (including percentages, currency, units of measurement etc), print the numeric answer in words. For example, if the answer is '32' print 'thirty-two' without quotes. Do not print the option label. Think step by step and in the end, finish your response with 'Response:$RESPONSE' where $RESPONSE (without quotes) is the final output expected.

**Question:** How do I make the pattern for the baby leather shoes?

**Options:**
A. Create a template on a piece of paper by placing your babies shoe on the paper and drawing around it.
B. Create a template on a piece of paper by placing your babies foot on the paper and drawing around it.

**Ground Truth:** A

**Instruction Output:** Response:Create a template on a piece of paper by placing your babies shoe on the paper and drawing around it.

Given a passage and a boolean question, and the possible answer candidates 'A' or 'B', print the list of incorrect answers (not the answer label). Increase each value by 1 while printing if it is a numeric quanity including dollar values, percentages but contains no other string or units of measurement. Do not print the option labels. Think step by step and in the end, finish your response with 'Response:$RESPONSE' where $RESPONSE (without quotes) is the final output expected.

Passage: A Star Is Born is an upcoming American musical romantic drama film produced and directed by Bradley Cooper, in his directorial debut. Cooper also wrote the screenplay with Will Fetters and Eric Roth. A remake of the 1937 film of the same name, it stars Cooper, Lady Gaga, Andrew Dice Clay, Dave Chappelle, and Sam Elliott, and follows a hard-drinking country musician (Cooper) who discovers and falls in love with a young singer (Gaga). It marks the third remake of the original 1937 film (which featured Janet Gaynor and Fredric March), which was adapted into a 1954 musical (starring Judy Garland and James Mason) and then remade as a 1976 rock musical with Barbra Streisand and Kris Kristofferson.

**Question:** is bradley cooper a star is born a remake

**Options:**
A. True
B. False

**Ground Truth:** A
**Instruction Output:** Response:['False']

---

Dataset: PIQA
Instruction: capitalize_correct_answer

Given a question and two answer candidates 'A' and 'B', answer the question by printing the text associated with the correct option label in uppercase. Do not print the option label. Think step by step and in the end, finish your response with 'Response:$RESPONSE' where $RESPONSE (without quotes) is the final output expected.

**Question:** wool

**Options:**
A. can be used to line cookie tins
B. can be used to line pants

**Ground Truth:** B
**Instruction Output:** Response:CAN BE USED TO LINE PANTS

## B.5 Metrics and Output Post-processing

We report the model performance as a form of exact match (EM) where we perform basic string parsing (removing beginning and ending whitespaces, quotations, etc.) and compare the model prediction to the expected output for the applied instruction.

Further, all our task prompt templates, as shown in Appendix B.4, explicitly instruct the model to provide their final response after a 'Response:' keyword. We search for and extract the response after this keyword while computing the exact match score. However, we observe that models may not always follow this, and can instead generate a wide range of other keywords (e.g. ⟨*the final answer is*, *the output is*, *etc*⟩, or no keyword at all. Given the diverse possible responses, we make a good-faith attempt to capture these patterns as part of our evaluation to classify a wider range of model responses.[8] As a result, we do not penalize models for not following this output format requirement.

However, models do still make errors when following the primary instruction. These could be minor copying errors, such as missing a period or comma, or even fixing typographical errors within the provided options. On the other hand, they could also be instruction following mistakes, where for instance, the option label is added to the response even when the prompt explicitly states otherwise.

---

[8]We find that more than 94% of outputs across all models and samples contained the 'Response:' string

We do not expect models to make such mistakes given clear instructions, and therefore, penalize models for these types of errors in our evaluation.

## B.6 Qualitative examples of error types

### B.6.1 Type 1: Instruction following error

Model: Llama-3.1-8B-Instruct

Input Task (Append correct answer with a string):

```
Given a sentence with a blank and 2 options namely 'A' and 'B', print the text associated with the option
    label that answers the question correctly. Do not print the option label. Without including any
    additional characters, also append the following text string at the end of the output: 'Date'. Do not
    add any special characters including quotations, spaces, new lines etc in the response. Think step
    by step and in the end, finish your response with 'Response:$RESPONSE' where $RESPONSE (without
    quotes) is the final output expected.

Sentence: Samantha strictly followed their diet while Sarah ate what they felt like, and _ ate chicken and
    broccoli all week.

Options: A. Samantha B. Sarah
```

Ground Truth:

```
SamanthaDate
```

Output:

```
Date
Response:Samantha
```

### B.6.2 Type 2: Knowledge Error

Model: Qwen 2.5-72B-instruct

Input Task (Print the correct answer text:

```
Given a question about biology and 10 options: A, B, C, D, E, F, G, H, I, J as candidate answers, print
    the text associated with the option label that answers the question correctly. Do not print the
    option label. Think step by step and in the end, finish your response with 'Response:$RESPONSE'
    where $RESPONSE (without quotes) is the final output expected.

Question: Nymphaea odorata, a water lily, is an aquatic plant whose leaves and flowers float on the surface of
    the water. Which of the following characteristics would be expected for leaves of this plant?
A. Stomata on the bottom surface with no water-conducting (xylem) cells
B. stomata on the bottom surface and relatively few water-conducting (xylem) cells
C. Stomata on the top surface with no water-conducting (xylem) cells
D. Stomata evenly distributed on both surfaces and relatively many water-conducting (xylem) cells
E. stomata on the top surface and relatively few water-conducting (xylem) cells
F. No stomata and relatively many water-conducting (xylem) cells
G. stomata on the top surface and relatively many water-conducting (xylem) cells
H. No stomata and relatively few water-conducting (xylem) cells
I. Stomata evenly distributed on both surfaces and relatively few water-conducting (xylem) cells
J. stomata on the bottom surface and relatively many water-conducting (xylem) cells
```

Ground Truth:

```
    stomata on the top surface and relatively few water-conducting (xylem) cells
```

Output:

```
    To determine the correct answer, let's analyze the characteristics of the leaves of Nymphaea odorata, a
        water lily.

1. **Stomata Location**:
    - Stomata are essential for gas exchange, which is crucial for photosynthesis and transpiration. In aquatic
        plants, especially those with floating leaves, stomata are typically located on the upper surface of
        the leaves. This is because the upper surface is exposed to the air, allowing for efficient gas
        exchange, while the lower surface is in contact with water, which would impede gas exchange.

2. **Water-Conducting Cells (Xylem)**:
    - Xylem is responsible for transporting water and minerals from the roots to the leaves. In aquatic plants,
        the availability of water is not a limiting factor, so the need for extensive xylem tissue is
        reduced. However, some xylem is still necessary to transport minerals and support the plant.

Given these points, we can eliminate options that do not align with these characteristics:
```

```
– Options A, B, and J, which suggest stomata on the bottom surface, are incorrect because the bottom surface
      is in contact with water, which would not allow for efficient gas exchange.
– Options C, E, and G, which suggest no stomata or stomata on the top surface, are more plausible. However, we
      need to consider the xylem content.
– Options F and H, which suggest no stomata, are incorrect because stomata are necessary for gas exchange.
– Options D and I, which suggest stomata evenly distributed on both surfaces, are less likely because the
      bottom surface is in contact with water, making stomata on the bottom less effective.

The most plausible option is:
– **G. stomata on the top surface and relatively many water-conducting (xylem) cells**

This option aligns with the need for stomata on the top surface for gas exchange and the presence of some
      xylem for mineral transport and support.

Response: stomata on the top surface and relatively many water-conducting (xylem) cells
```

### B.6.3   Type 3: Both Knowledge and Instruction following errors

Model: Gemma-2-9b-it

Input Task - Capitalize correct answer:

```
Given a sentence with a blank and 2 options namely 'A' and 'B', capitalize the text associated with the
      optional label that answers the question correctly. Do not print the option label. Think step by step
      and in the end, finish your response with 'Response:$RESPONSE' where $RESPONSE (without quotes) is the
      final output expected.

Sentence: John could not hear his alarm clock when he was sleeping with a headphone on his head because the \_
      is closer. Options: A. headphone B. clock
```

Ground truth:

```
HEADPHONE
```

Model Output:

```
Let's analyze the sentence step by step:
– The sentence is about John not being able to hear his alarm clock.
– The blank is where we need to insert either 'headphone' or 'clock'.
– The sentence states that John is wearing headphones on his head.
– Therefore, the alarm clock would be closer to his head than the headphones.
– This means the alarm clock would be the closer object.
– The correct option to fill in the blank is 'clock'.
Now, let's capitalize the text associated with the correct option.
Response:clock
```

### B.6.4   Type 4: Unclassified Errors

Model: Qwen 2.5-3B-instruct

Input Task (Print the correct answer text:

```
Given a mathematical question and 5 options namely 'a', 'b', 'c', 'd', and, 'e', as candidate answers,
      print the text associated with the option label that answers the question correctly. Do not print
      the option label. Without including any additional characters, also append the following text string
      at the end of the output:'xQbkTIPMZ7wlBJCx'. Do not add any special characters including quotations
      , spaces, new lines etc in the response. Think step by step and in the end, finish your response
      with 'Response:$RESPONSE' where $RESPONSE (without quotes) is the final output expected.

Question: two friends c and d leave point c and point d simultaneously and travel towards point d and point c
      on the same route at their respective constant speeds. they meet along the route and immediately
      proceed to their respective destinations in 32 minutes and 50 minutes respectively. how long will d
      take to cover the entire journey between point d and point c ?
Options:
a. 90
b. 80
c. 75
d. 60
e. 65
```

Output:

```
To solve this problem, let's denote the distance between point C and point D as \(D\), the speed of friend
      C as \(v_C\), and the speed of friend D as \(v_D\). Let's also denote the time they meet as \(t\)
      minutes.

When they meet, the sum of the distances they have traveled is equal to the total distance \(D\). So, we have:
\[ v_C t + v_D t = D \]
\[ t (v_C + v_D) = D \]
\[ t = \frac{D}{v_C + v_D} \]
```

```
After meeting, friend C takes 32 minutes to reach point D, and friend D takes 50 minutes to reach point C.
    This means:
\[ \frac{D - v_C t}{v_C} = 32 \]
\[ \frac{D - v_D t}{v_D} = 50 \]

Substituting \( t = \frac{D}{v_C + v_D}\) into the equations:
\[ \frac{D - v_C \frac{D}{v_C + v_D}}{v_C} = 32 \]
\[ \frac{D - v_D \frac{D}{v_C + v_D}}{v_D} = 50 \]

Simplifying these equations:
\[ \frac{D (1 - \frac{v_C}{v_C + v_D})}{v_C} = 32 \]
\[ \frac{D (1 - \frac{v_D}{v_C + v_D})}{v_D} = 50 \]

\[ \frac{D \frac{v_D}{v_C + v_D}}{v_C} = 32 \]
\[ \frac{D \frac{v_C}{v_C + v_D}}{v_D} = 50 \]

\[ \frac{D v_D}{v_C (v_C + v_D)} = 32 \]
\[ \frac{D v_C}{v_D (v_C + v_D)} = 50 \]
..... continued incoherent reasoning and no final response provided
```

## B.7  Additional Benchmark Statistics

The following sections reports detailed statistics for the Full and Lite Benchmark. We report statistics for both instruction following and Instructions with no-effect subsets. We observe that for some dataset (knowledge tasks) and instruction combinations, the corresponding entries are zero indicating that there is no single instance where the instruction gets applied (Instructions with no-effect) or there is no single instance where the instruction doesn't get applied (instruction follow subset).

Table 6: Full Benchmark: Instruct Follow Stats

| | BoolQ | MMLUPro | | | | | | | | | | | | | | PIQA | MathQA | Winogrande |
| --- | --- | --- | --- | --- | --- | --- | --- | --- | --- | --- | --- | --- | --- | --- | --- | --- | --- | --- |
| | | Physics | Health | Economics | Law | Philosophy | Business | Other | Chemistry | Psychology | History | Computer Science | Biology | Math | Engineering | | | |
| numformat_numeric_answer | 0 | 150 | 53 | 150 | 150 | 44 | 150 | 150 | 150 | 150 | 47 | 150 | 150 | 150 | 150 | 553 | 1500 | 0 |
| increment_incorrect_numeric_answers_by_one | 1500 | 150 | 150 | 150 | 0 | 0 | 150 | 150 | 150 | 150 | 0 | 150 | 0 | 150 | 150 | 1500 | 1500 | 1267 |
| sort_only_incorrect_answers | 1500 | 150 | 150 | 150 | 150 | 150 | 150 | 150 | 150 | 150 | 150 | 150 | 150 | 150 | 150 | 1500 | 1500 | 1267 |
| use_options_to_create_string | 1500 | 150 | 150 | 150 | 150 | 150 | 150 | 150 | 150 | 150 | 150 | 150 | 150 | 150 | 150 | 1500 | 1500 | 1267 |
| print_correct_answer_label | 1500 | 150 | 150 | 150 | 150 | 150 | 150 | 150 | 150 | 150 | 150 | 150 | 150 | 150 | 150 | 1500 | 1500 | 1267 |
| print_correct_answer | 1500 | 150 | 150 | 150 | 150 | 150 | 150 | 150 | 150 | 150 | 150 | 150 | 150 | 150 | 150 | 1500 | 1500 | 1267 |
| reverse_correct_answer_alternate_case | 1500 | 150 | 150 | 150 | 150 | 150 | 150 | 150 | 150 | 150 | 150 | 150 | 150 | 150 | 150 | 1500 | 540 | 1267 |
| increment_correct_numeric_answer_by_one | 0 | 150 | 22 | 42 | 1 | 4 | 150 | 150 | 29 | 11 | 1 | 86 | 13 | 150 | 53 | 0 | 0 | 0 |
| alternate_case_correct_answer | 1500 | 150 | 150 | 150 | 150 | 150 | 150 | 150 | 150 | 150 | 150 | 150 | 150 | 150 | 150 | 1500 | 1500 | 1267 |
| print_correct_answer_append_string | 1500 | 150 | 150 | 150 | 150 | 150 | 150 | 150 | 150 | 150 | 150 | 150 | 150 | 150 | 150 | 1500 | 1500 | 1267 |
| print_correct_answer_in_words | 0 | 142 | 14 | 10 | 0 | 3 | 81 | 31 | 108 | 10 | 0 | 84 | 8 | 150 | 35 | 0 | 1500 | 0 |
| sort_options_to_create_string | 1500 | 150 | 150 | 150 | 150 | 150 | 150 | 150 | 150 | 150 | 150 | 150 | 150 | 150 | 150 | 1500 | 1500 | 1267 |
| reverse_correct_answer | 1500 | 150 | 150 | 150 | 150 | 150 | 150 | 150 | 150 | 150 | 150 | 150 | 150 | 150 | 150 | 1500 | 1500 | 1267 |
| use_incorrect_options_to_create_string | 1500 | 150 | 150 | 150 | 150 | 150 | 150 | 150 | 150 | 150 | 150 | 150 | 150 | 150 | 150 | 1500 | 1500 | 1267 |
| capitalize_correct_answer | 1500 | 150 | 150 | 150 | 150 | 150 | 150 | 150 | 150 | 150 | 150 | 150 | 150 | 150 | 150 | 1500 | 1500 | 1267 |

Table 7: Full Benchmark: Instructions with no-effect

| | MMLUPro | | | | | | | | | | | | | | MathQA | PIQA | BoolQ | Winogrande |
| --- | --- | --- | --- | --- | --- | --- | --- | --- | --- | --- | --- | --- | --- | --- | --- | --- | --- | --- |
| | Health | Economics | Math | Psychology | Law | Computer Science | Physics | Other | Business | Chemistry | Engineering | Biology | History | Philosophy | | | | |
| numformat_numeric_answer | 150 | 150 | 150 | 150 | 150 | 150 | 150 | 150 | 150 | 150 | 150 | 150 | 150 | 150 | 1337 | 1285 | 1500 | 1267 |
| print_correct_answer_in_words | 150 | 150 | 150 | 150 | 150 | 150 | 150 | 150 | 150 | 150 | 150 | 150 | 150 | 150 | 1331 | 1500 | 1500 | 1267 |
| increment_correct_numeric_answer_by_one | 150 | 150 | 150 | 150 | 150 | 150 | 150 | 150 | 150 | 150 | 150 | 150 | 150 | 150 | 928 | 1500 | 1500 | 1267 |
| reverse_correct_answer_alternate_case | 30 | 48 | 150 | 13 | 3 | 122 | 150 | 150 | 150 | 150 | 150 | 33 | 2 | 15 | 1500 | 0 | 0 | 0 |
| reverse_correct_answer | 0 | 0 | 0 | 0 | 0 | 0 | 0 | 0 | 0 | 0 | 0 | 0 | 0 | 0 | 1 | 0 | 1 | 0 |
| use_incorrect_options_to_create_string | 0 | 0 | 0 | 0 | 0 | 0 | 0 | 0 | 0 | 0 | 0 | 0 | 0 | 0 | 0 | 1 | 0 | 0 |

Table 8: Lite Benchmark: Instruct Follow Stats

| | MMLUPro | | | | | | | | | | | | | | | | | |
|---|---|---|---|---|---|---|---|---|---|---|---|---|---|---|---|---|---|---|
| | BoolQ | chemistry | other | physics | math | biology | philosophy | psychology | economics | history | health | law | engineering | business | computer science | PIQA | MathQA | Winogrande |
| print_correct_answer | 150 | 66 | 82 | 65 | 53 | 53 | 50 | 54 | 69 | 49 | 60 | 38 | 74 | 73 | 54 | 253 | 352 | 150 |
| print_correct_answer_label | 150 | 66 | 82 | 65 | 53 | 53 | 50 | 54 | 69 | 49 | 60 | 38 | 74 | 73 | 54 | 253 | 352 | 150 |
| increment_correct_numeric_answer_by_one | 0 | 25 | 25 | 25 | 25 | 13 | 4 | 11 | 25 | 1 | 22 | 1 | 25 | 25 | 25 | 0 | 150 | 0 |
| sort_options_to_create_string | 150 | 25 | 25 | 25 | 25 | 25 | 25 | 25 | 25 | 25 | 25 | 25 | 25 | 25 | 25 | 150 | 150 | 150 |
| print_correct_answer_in_words | 0 | 25 | 25 | 25 | 25 | 8 | 3 | 10 | 10 | 0 | 14 | 0 | 25 | 25 | 25 | 0 | 150 | 0 |
| reverse_correct_answer | 150 | 25 | 25 | 25 | 25 | 25 | 25 | 25 | 25 | 25 | 25 | 25 | 25 | 25 | 25 | 150 | 150 | 150 |
| use_incorrect_options_to_create_string | 150 | 25 | 25 | 25 | 25 | 25 | 25 | 25 | 25 | 25 | 25 | 25 | 25 | 25 | 25 | 150 | 150 | 150 |
| use_options_to_create_string | 150 | 25 | 25 | 25 | 25 | 25 | 25 | 25 | 25 | 25 | 25 | 25 | 25 | 25 | 25 | 150 | 150 | 150 |
| print_correct_answer_append_string | 150 | 25 | 25 | 25 | 25 | 25 | 25 | 0 | 25 | 0 | 25 | 25 | 25 | 25 | 25 | 150 | 150 | 150 |
| increment_incorrect_numeric_answers_by_one | 150 | 25 | 25 | 25 | 25 | 25 | 25 | 25 | 25 | 25 | 25 | 25 | 25 | 25 | 25 | 150 | 150 | 150 |
| sort_only_incorrect_answers | 150 | 25 | 25 | 25 | 25 | 0 | 0 | 0 | 25 | 0 | 25 | 0 | 25 | 25 | 25 | 150 | 150 | 150 |
| numformat_numeric_answer | 0 | 25 | 25 | 25 | 25 | 25 | 25 | 25 | 25 | 25 | 25 | 25 | 25 | 25 | 25 | 150 | 150 | 0 |
| reverse_correct_answer_alternate_case | 150 | 25 | 25 | 25 | 25 | 25 | 25 | 25 | 25 | 25 | 25 | 25 | 25 | 25 | 25 | 150 | 150 | 150 |
| alternate_case_correct_answer | 150 | 25 | 25 | 25 | 25 | 25 | 25 | 25 | 25 | 25 | 25 | 25 | 25 | 25 | 25 | 150 | 150 | 150 |
| capitalize_correct_answer | 150 | 25 | 25 | 25 | 25 | 25 | 25 | 25 | 25 | 25 | 25 | 25 | 25 | 25 | 25 | 150 | 150 | 150 |

Table 9: Lite Benchmark: Instructions with no-effect

| | MMLUPro | | | | | | | | | | | | | | | | | |
|---|---|---|---|---|---|---|---|---|---|---|---|---|---|---|---|---|---|---|
| | biology | health | law | engineering | chemistry | math | business | physics | history | psychology | other | computer science | economics | philosophy | MathQA | Winogrande | BoolQ | PIQA |
| print_correct_answer | 57 | 47 | 41 | 56 | 46 | 51 | 48 | 52 | 29 | 44 | 50 | 63 | 57 | 43 | 293 | 150 | 150 | 198 |
| print_correct_answer_label | 57 | 47 | 41 | 56 | 46 | 51 | 48 | 52 | 29 | 44 | 50 | 63 | 57 | 43 | 293 | 150 | 150 | 198 |
| reverse_correct_answer_alternate_case | 25 | 25 | 3 | 25 | 25 | 25 | 25 | 25 | 2 | 13 | 25 | 25 | 25 | 15 | 150 | 0 | 0 | 0 |
| print_correct_answer_in_words | 25 | 25 | 25 | 25 | 25 | 25 | 25 | 25 | 25 | 25 | 25 | 25 | 25 | 25 | 150 | 150 | 150 | 150 |
| increment_correct_numeric_answer_by_one | 25 | 25 | 25 | 25 | 25 | 25 | 25 | 25 | 25 | 25 | 25 | 25 | 25 | 25 | 150 | 150 | 150 | 150 |
| numformat_numeric_answer | 25 | 25 | 25 | 25 | 25 | 25 | 25 | 25 | 25 | 25 | 25 | 25 | 25 | 25 | 150 | 150 | 150 | 150 |
| reverse_correct_answer | 0 | 0 | 0 | 0 | 0 | 0 | 0 | 0 | 0 | 0 | 0 | 0 | 0 | 0 | 1 | 0 | 0 | 0 |
| use_incorrect_options_to_create_string | 0 | 0 | 0 | 0 | 0 | 0 | 0 | 0 | 0 | 0 | 0 | 0 | 0 | 0 | 0 | 0 | 0 | 1 |

## C  Limitations

One of the primary limitations of our approach is that if a model is not skilled in the underlying knowledge/reasoning task, the study of instruction-following when conditioned on the model answer being correct isn't as helpful. Further, it is possible an LLM not considered in our set could generate responses in a style that is very incompatible with the assumptions made by our evaluation scripts. For example, some models could generate custom chain-of-thought markers which may require further adaptation of output processing and evaluation - we encourage users of our benchmark to inspect model outputs and update scripts if necessary.

## D  Additional Results

### D.1  Printing the correct answer

We present the comparison between model's performance on print correct answer and print correct answer labels tasks on the Lite Benchmark in Table 5. We observe that all models show a drop in performance when instructed to print correct answer instead of the label.

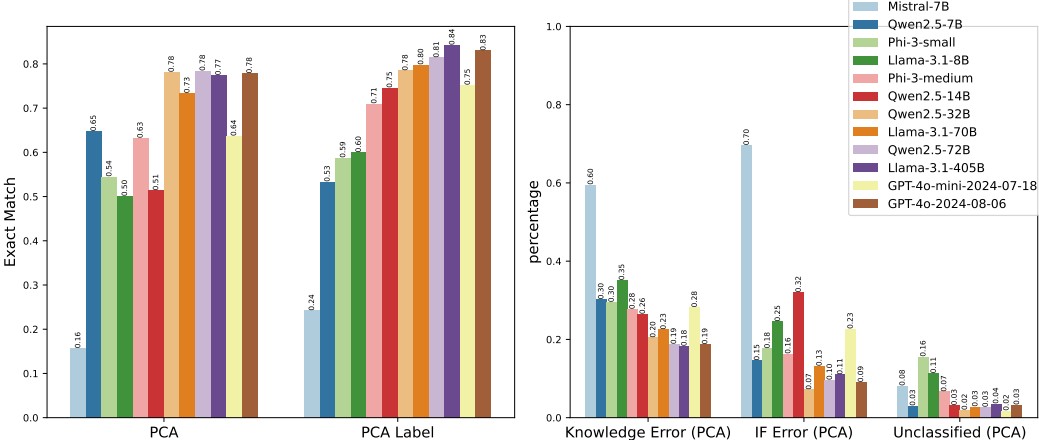

Figure 5: Lite Benchmark: Performance of LLMs on Printing the correct answer task and error comparison. PCA refers to *print_correct_answer* instruction and PCA label refers to *print_correct_answer_label*.

### D.2  Influence of Distractors

Figures 6, 7 and 8 show the performance of the Llama, Qwen2.5 and Phi family of models in the presence of distractor instructions. Figure 9 shows the performance of frontier models.

### D.3  Knowledge-Task Characteristics and Instruction-Following

We now present performance of different models for each instruction category in comparison with its corresponding performance on *print_correct_answer* (PCA). The patterns remains consistent.

**Numeric Manipulation Instructions:** Figures 10, 11 and 12 show the performance of small, medium and large models on the numeric manipulation category of instructions. These instructions only apply on the MathQA and MMLU datasets.

**String Manipulation Instructions:** Figures 14, 15, 16 and 17 show the performance of small, medium, large and frontier models on the string manipulation category of instructions. These instructions only apply on all datasets. Frontier models shown on data from Lite Benchmark.

**Format Correct Answer Instructions:** Figures 18, 19, 20 and 21 show the performance of small, medium, large and frontier models on the instructions that format the correct answer. These instructions only apply on all datasets. Frontier models shown on data from Lite Benchmark.

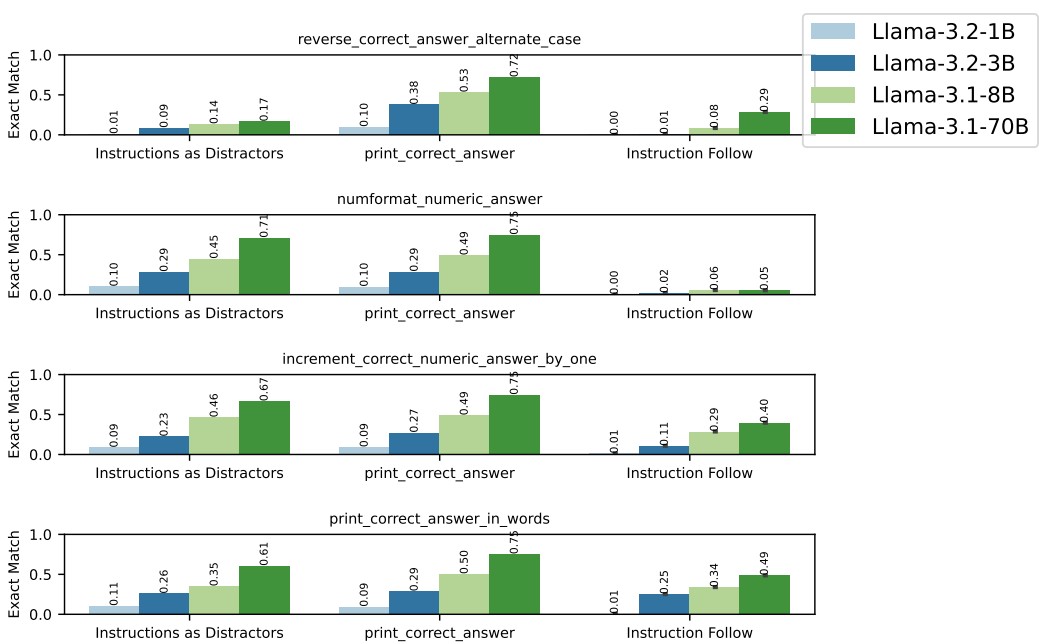

Figure 6: Distractors on Llama Family of Models

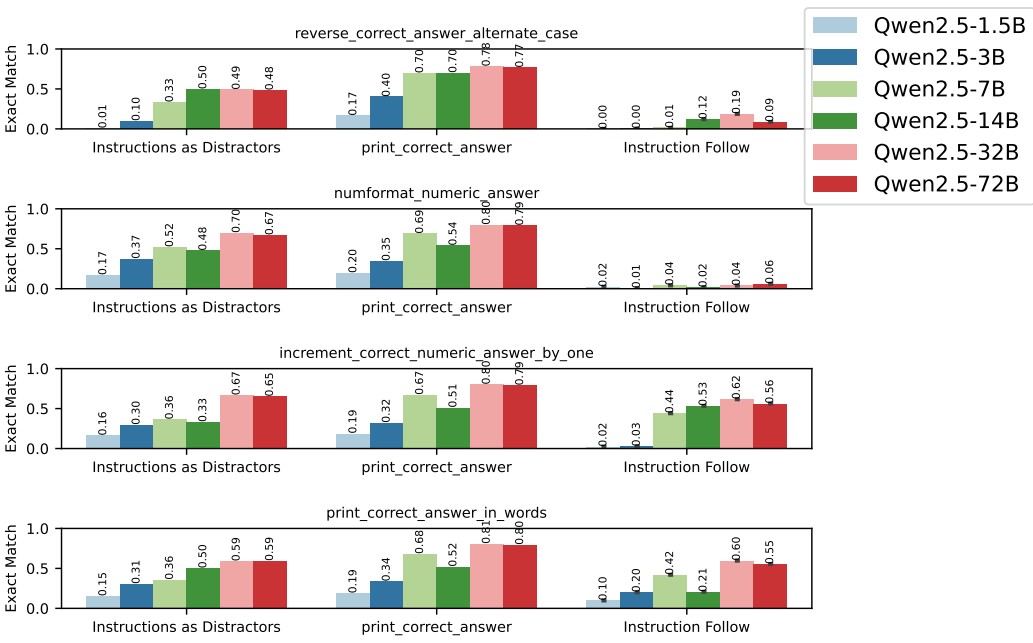

Figure 7: Distractors on Qwen Family of Models

**Operations on List:** Figures 22, 23, 24 and 25 show the performance of small, medium, large and frontier models on the instructions operate on lists. These instructions only apply on all datasets. Frontier models shown on data from Lite Benchmark.

**Operations on List (Conditional):** Figures 26, 27, 28 and 29 show the performance of small, medium, large and frontier models on the instructions operate on lists. These instructions only apply on all datasets. Frontier models shown on data from Lite Benchmark.

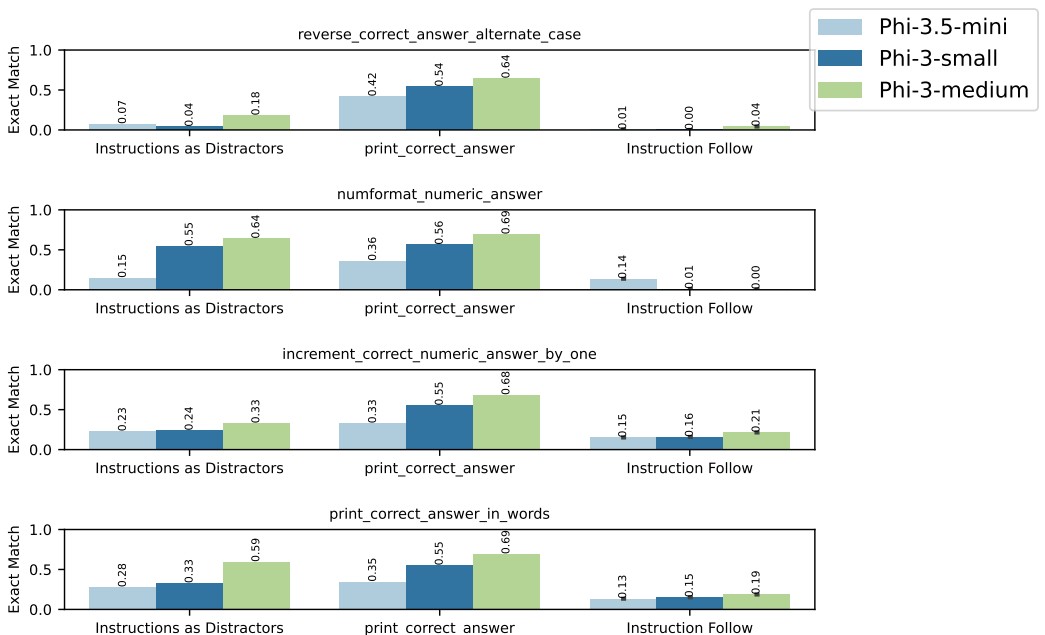

Figure 8: Distractors on Phi Family of Models

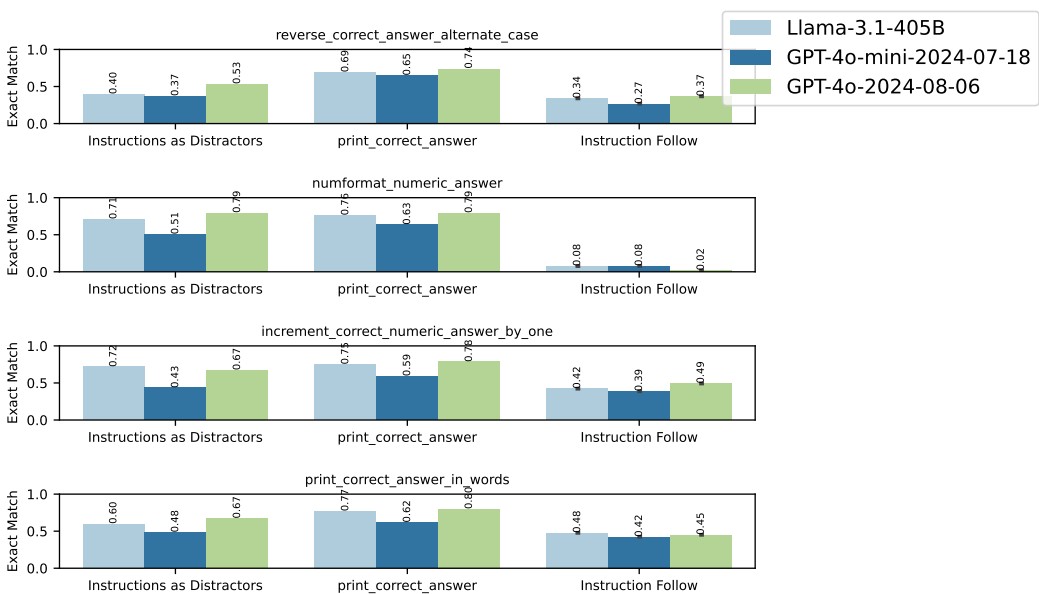

Figure 9: Distractors on Frontier Models

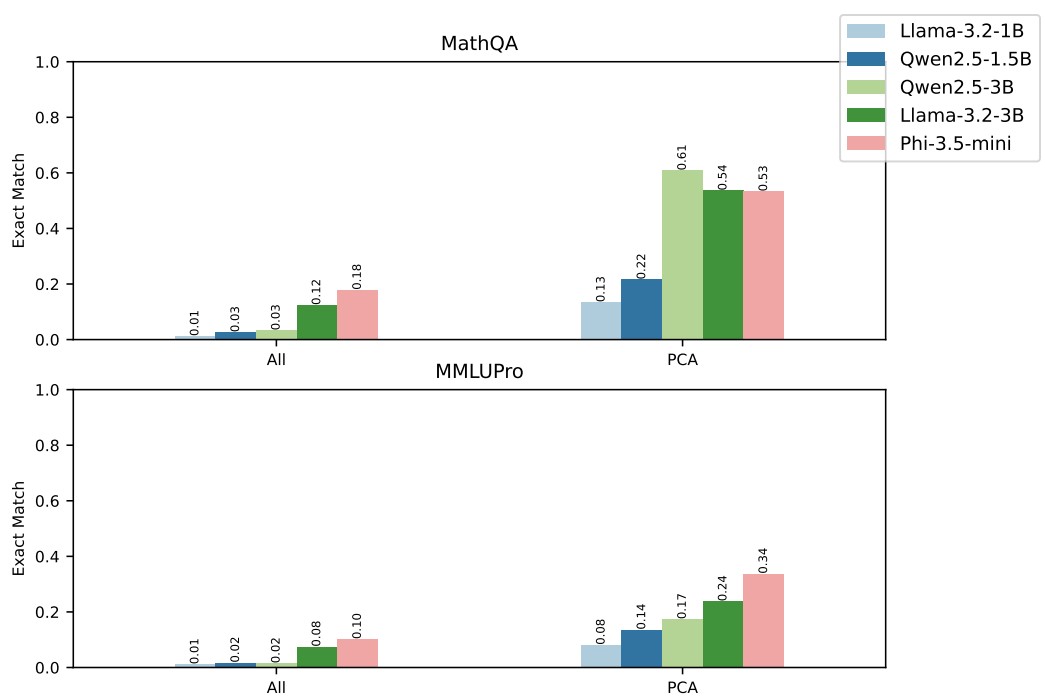

Figure 10: Small Scale Models: Performance variation of exact match scores for the Numeric Manipulation instruction category compared to its corresponding performance on $print\_correct\_answer$ (PCA).

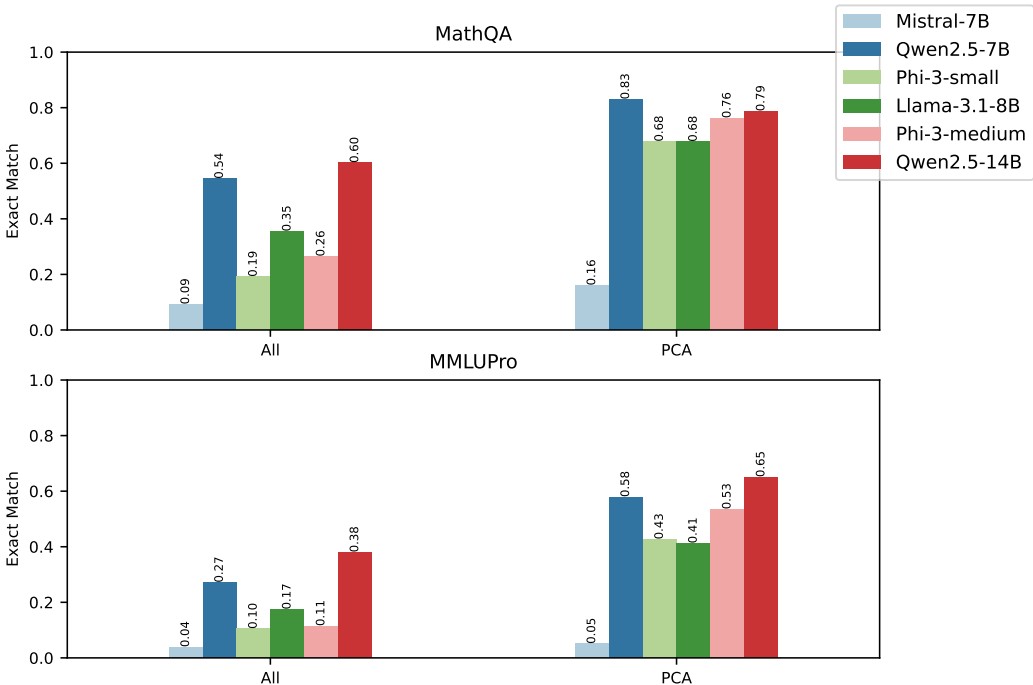

Figure 11: Medium Scale Models: Performance variation of exact match scores for the Numeric Manipulation instruction category compared to its corresponding performance on $print\_correct\_answer$ (PCA).

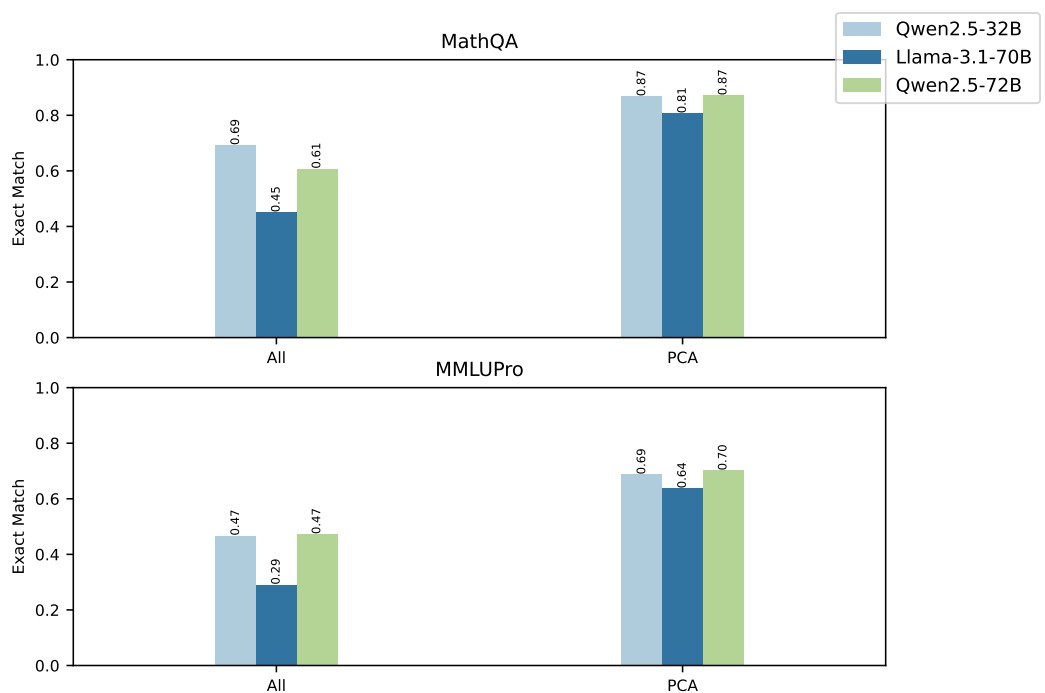

Figure 12: Large-Scale Models: Performance variation of exact match scores for the Numeric Manipulation instruction category compared to its corresponding performance on *print_correct_answer* (PCA).

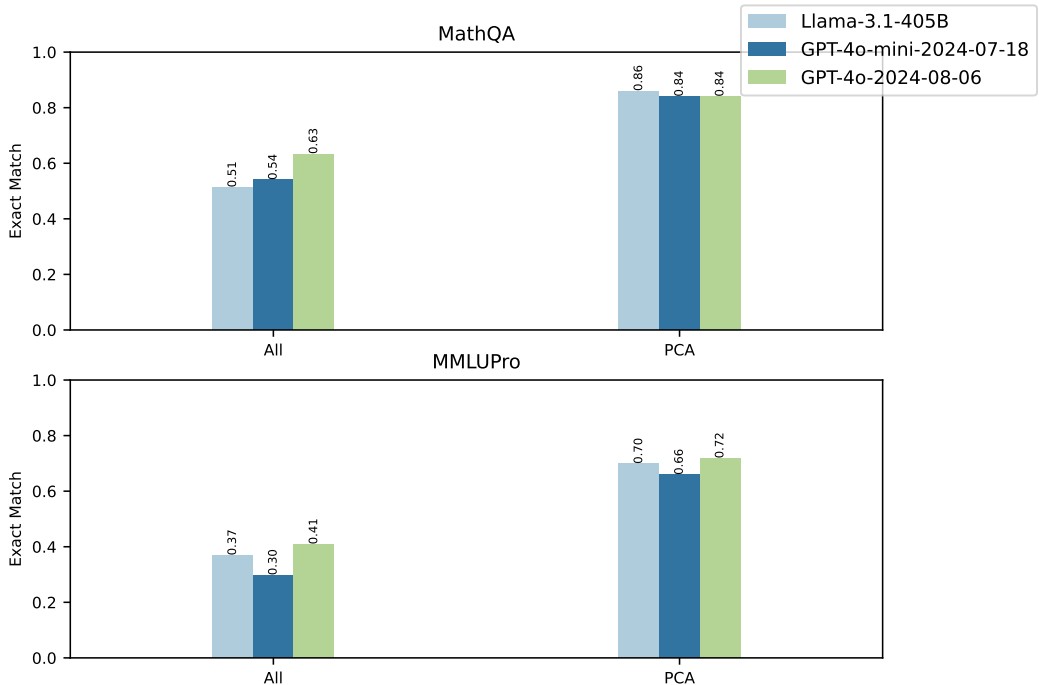

Figure 13: Frontier Models: Performance variation of exact match scores for the Numeric Manipulation instruction category compared to its corresponding performance on *print_correct_answer* (PCA).

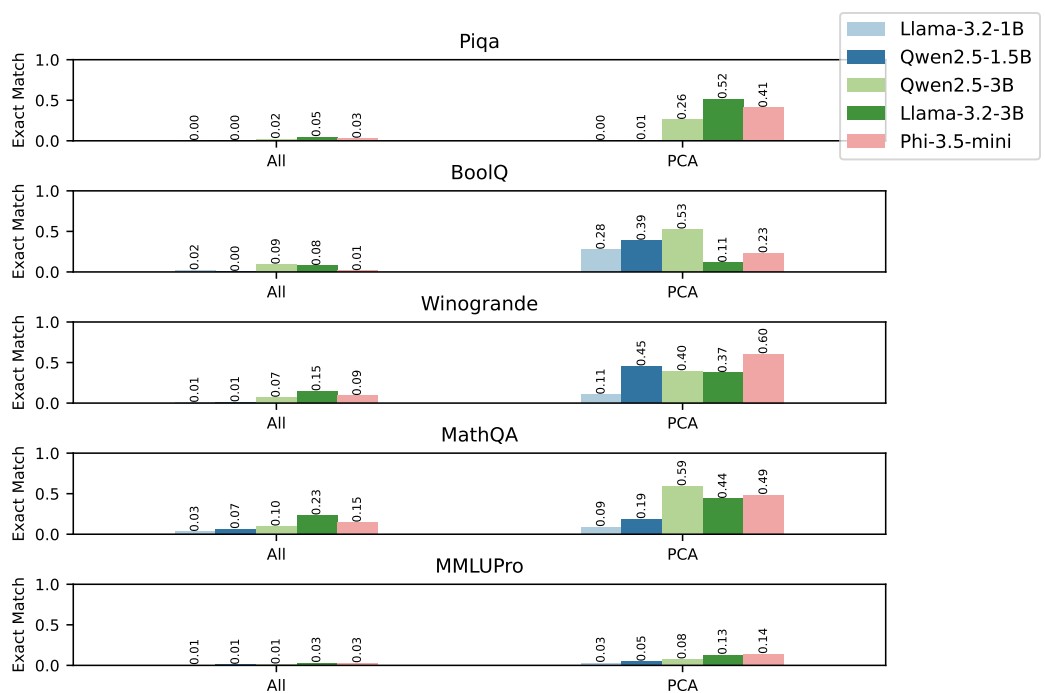

Figure 14: Small-Scale Models: Performance variation of exact match scores for the String Manipulation instruction category compared to its corresponding performance on $print\_correct\_answer$ (PCA).

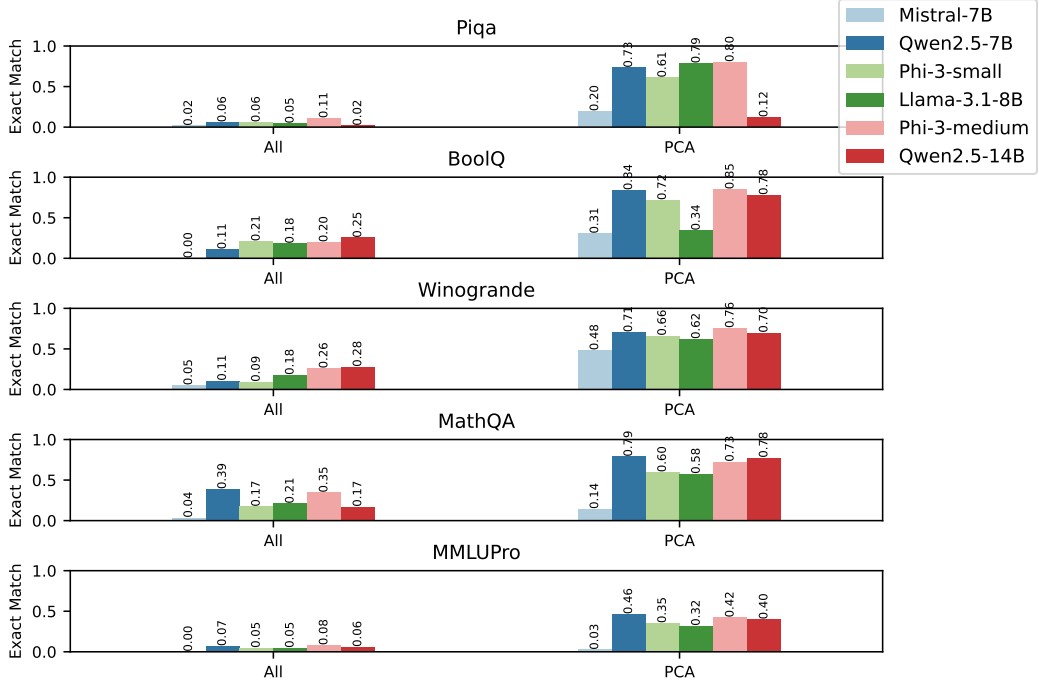

Figure 15: Medium-Scale Models: Performance variation of exact match scores for the String Manipulation instruction category compared to its corresponding performance on $print\_correct\_answer$ (PCA).

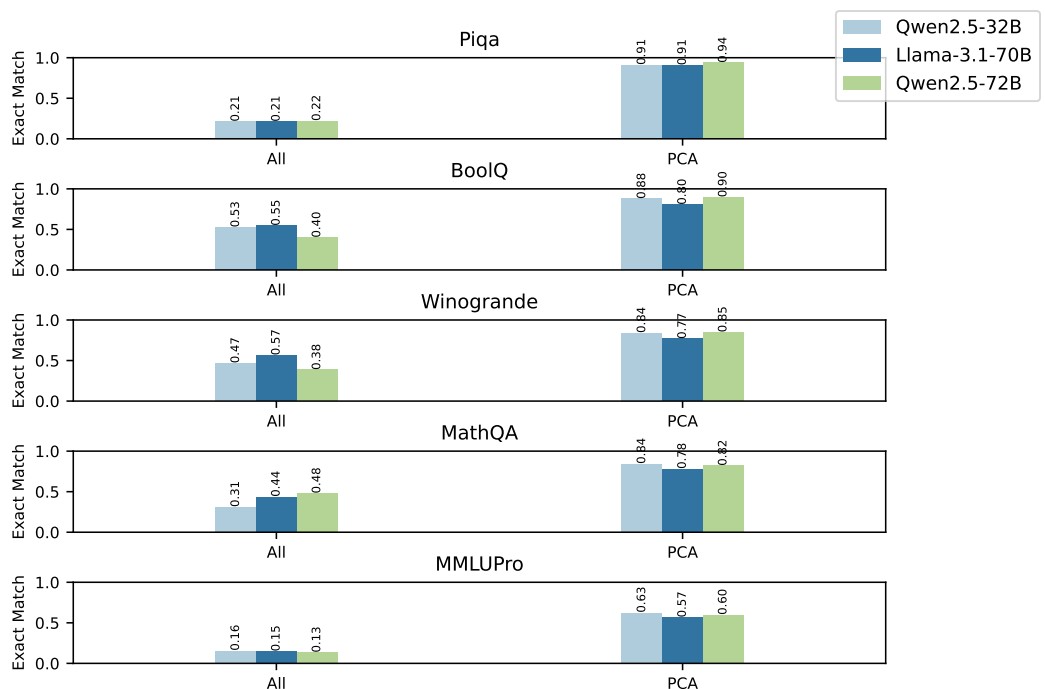

Figure 16: Large-Scale Models: Performance variation of exact match scores for the String Manipulation instruction category compared to its corresponding performance on *print_correct_answer* (PCA).

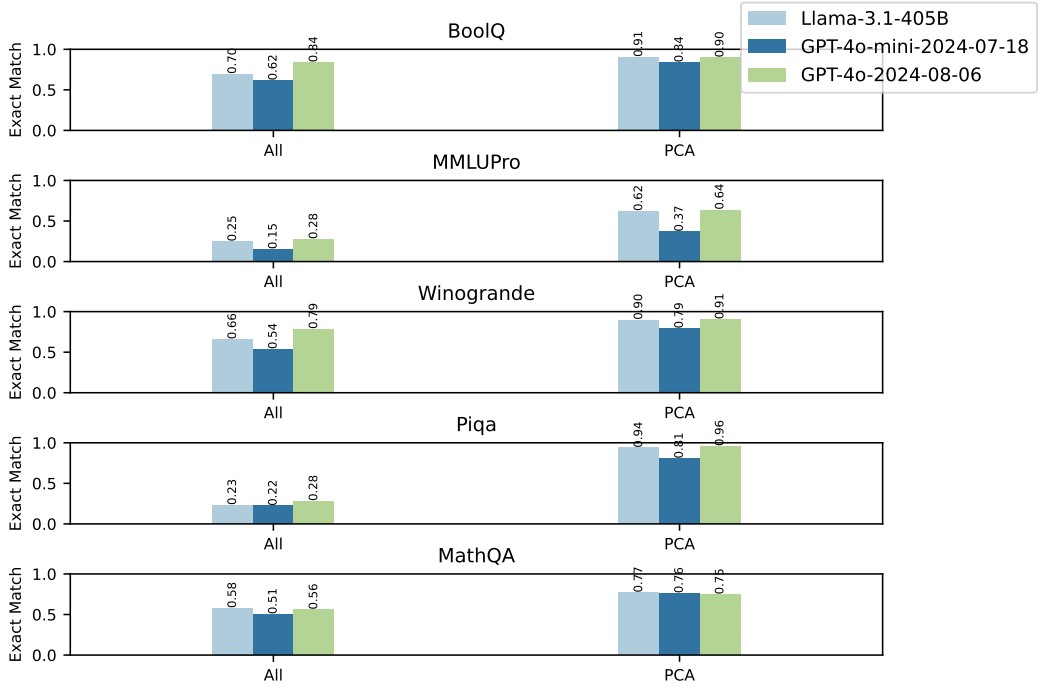

Figure 17: Frontier Models: Performance variation of exact match scores for the String Manipulation instruction category compared to its corresponding performance on *print_correct_answer* (PCA).

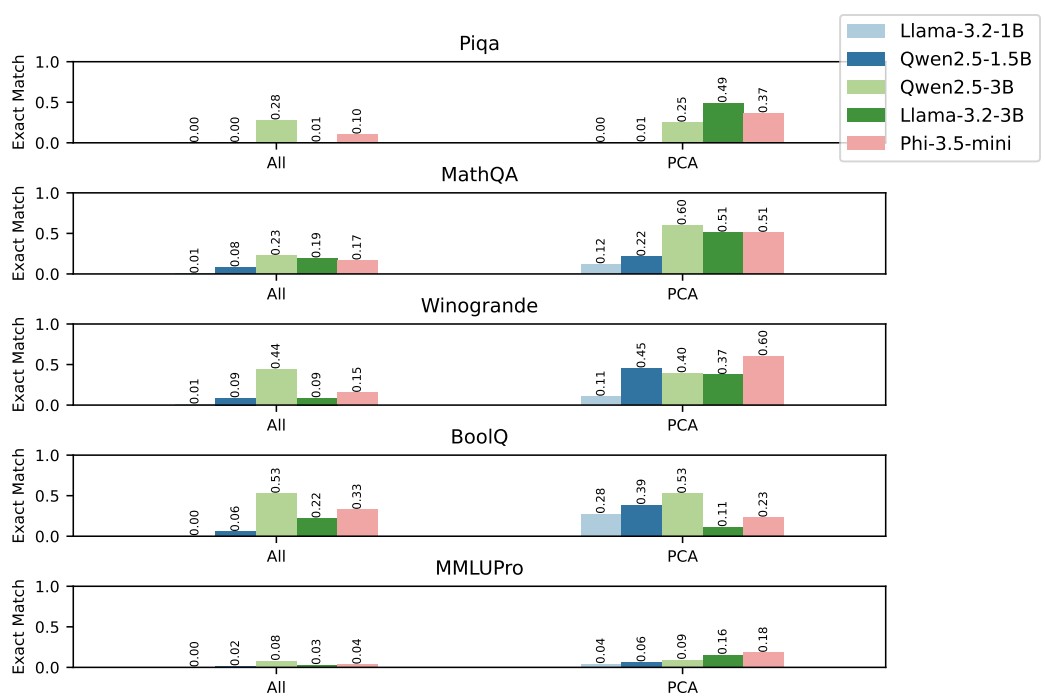

Figure 18: Small-Scale Models: Performance variation of exact match scores for the Format Correct Answer instruction category compared to its corresponding performance on $print\_correct\_answer$ (PCA).

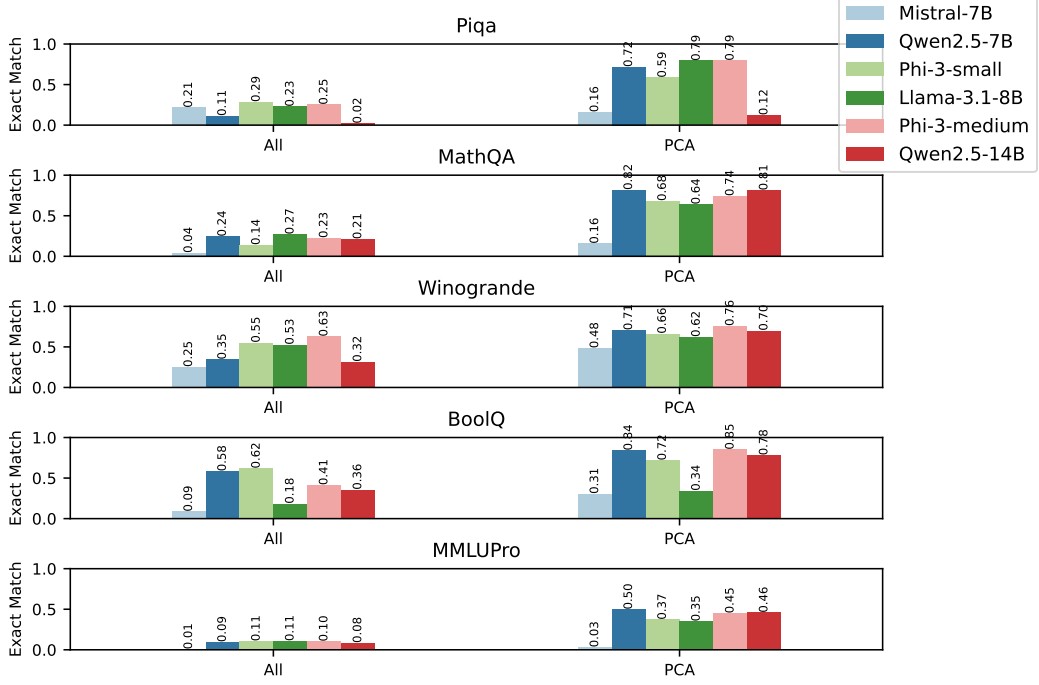

Figure 19: Medium-Scale Models: Performance variation of exact match scores for the Format Correct Answer instruction category compared to its corresponding performance on $print\_correct\_answer$ (PCA).

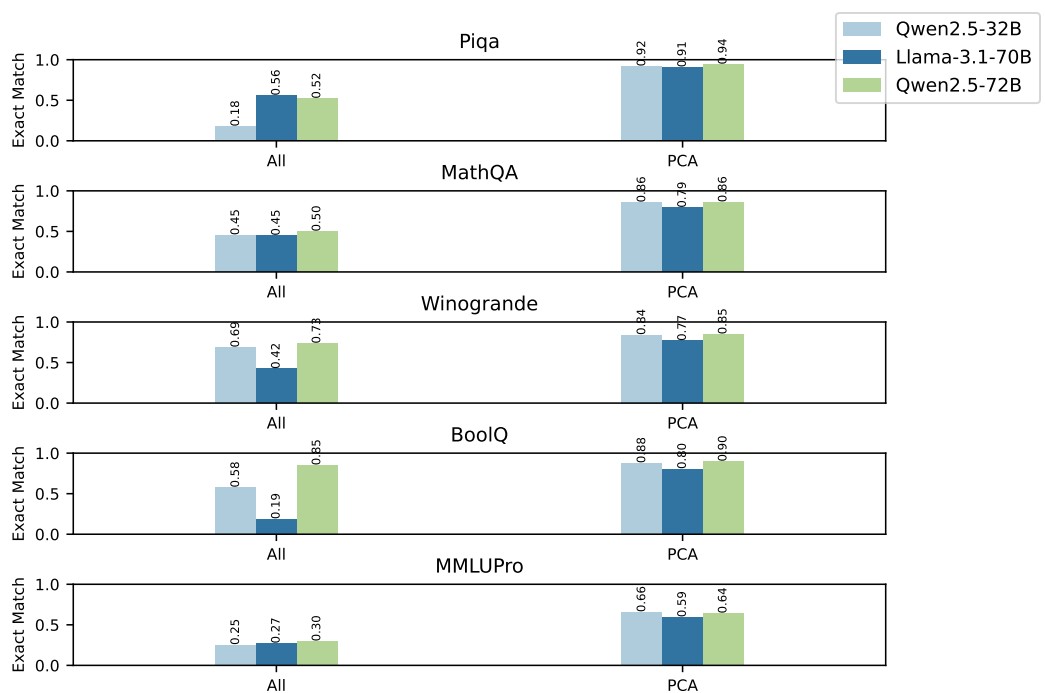

Figure 20: Large-Scale Models: Performance variation of exact match scores for the Format Correct Answer instruction category compared to its corresponding performance on *print_correct_answer* (PCA).

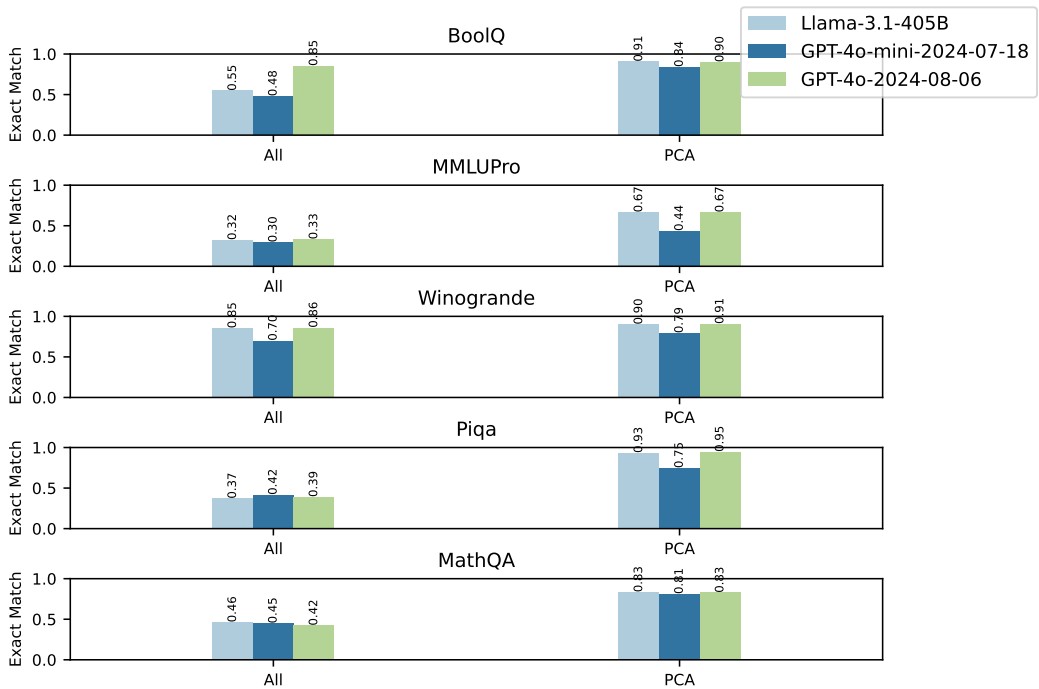

Figure 21: Frontier-Scale Models: Performance variation of exact match scores for the Format Correct Answer instruction category compared to its corresponding performance on *print_correct_answer* (PCA).

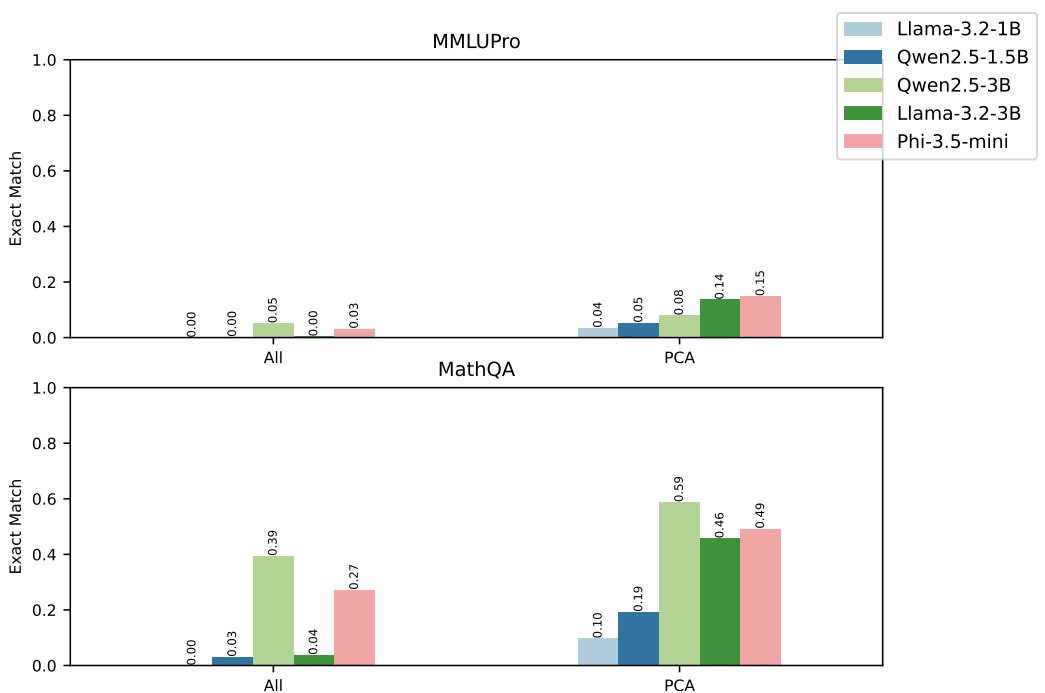

Figure 22: Small-Scale Models: Performance variation of exact match scores for the Operations on List instruction category compared to its corresponding performance on $print\_correct\_answer$ (PCA).

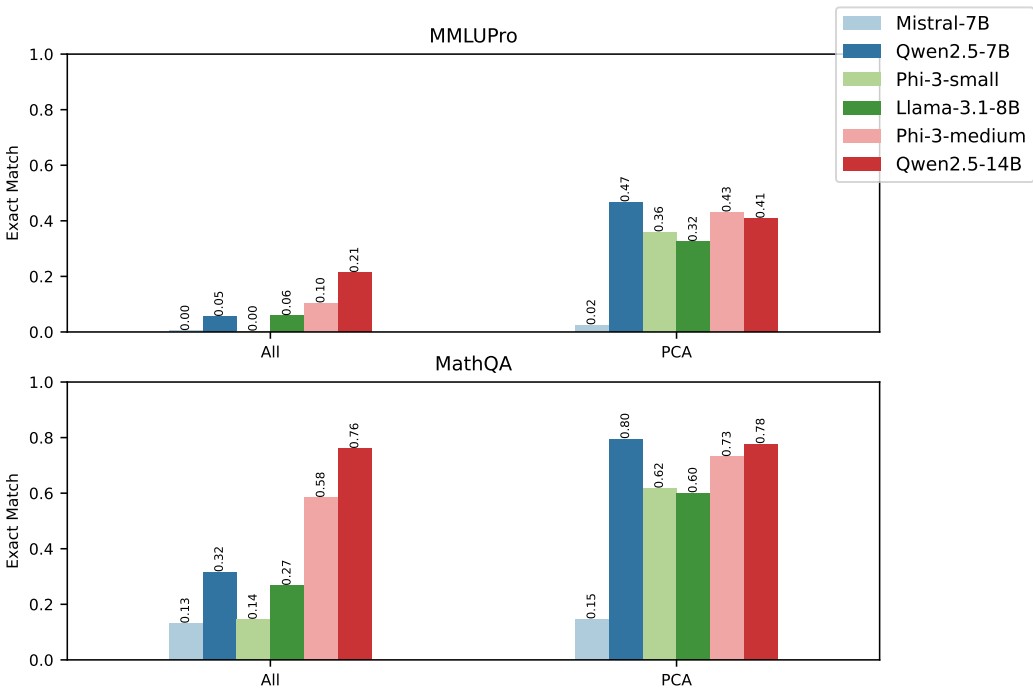

Figure 23: Medium-Scale Models: Performance variation of exact match scores for the Operations on List instruction category compared to its corresponding performance on $print\_correct\_answer$ (PCA).

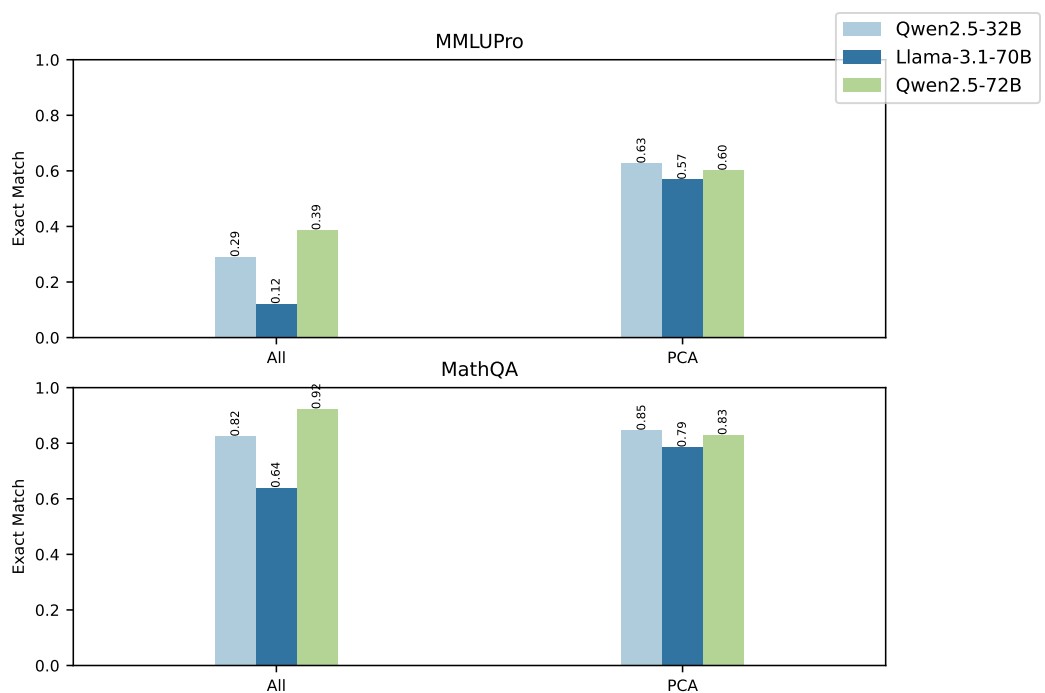

Figure 24: Large-Scale Models: Performance variation of exact match scores for the Operations on List instruction category compared to its corresponding performance on *print_correct_answer* (PCA).

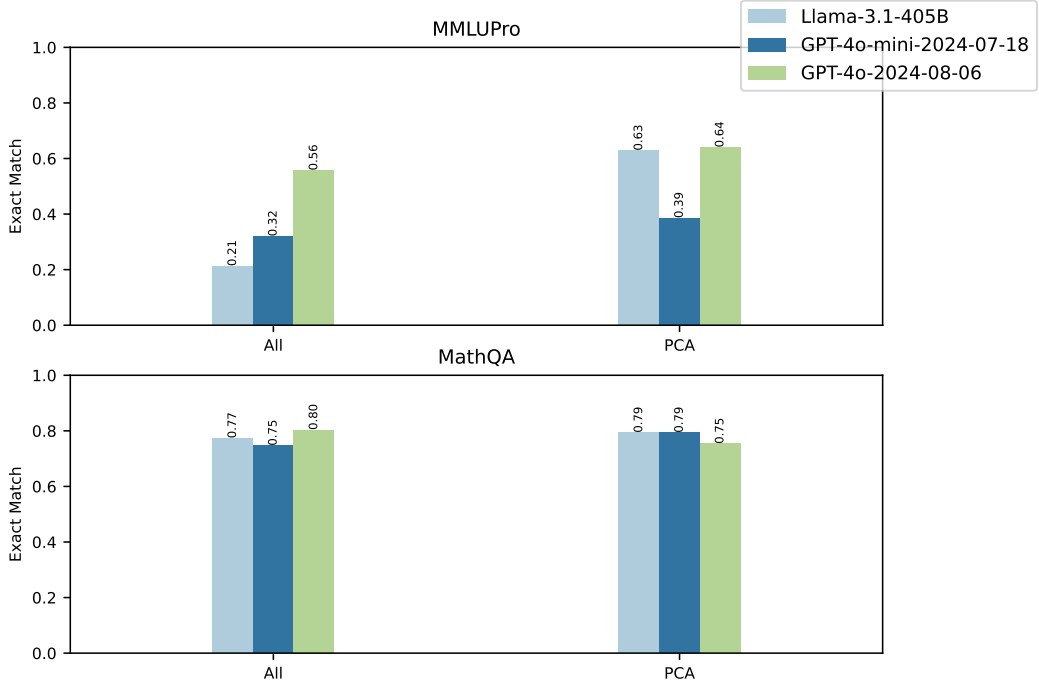

Figure 25: Frontier Models: Performance variation of exact match scores for the Operations on List instruction category compared to its corresponding performance on *print_correct_answer* (PCA).

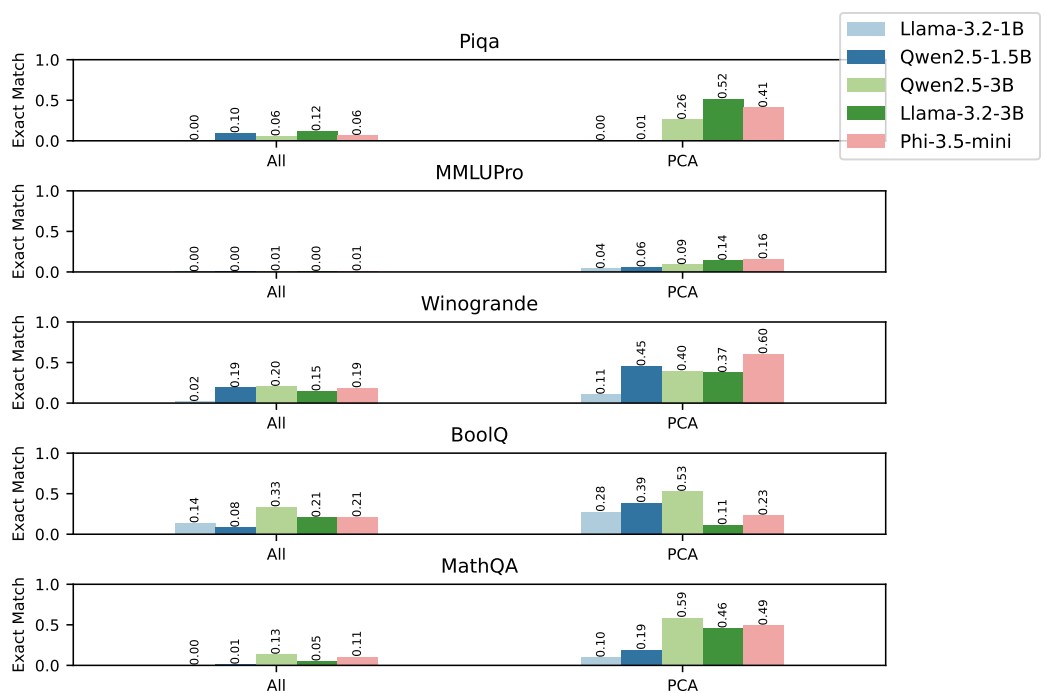

Figure 26: Small-Scale Models: Performance variation of exact match scores for the Operations on List (Conditional) instruction category compared to its corresponding performance on *print_correct_answer* (PCA).

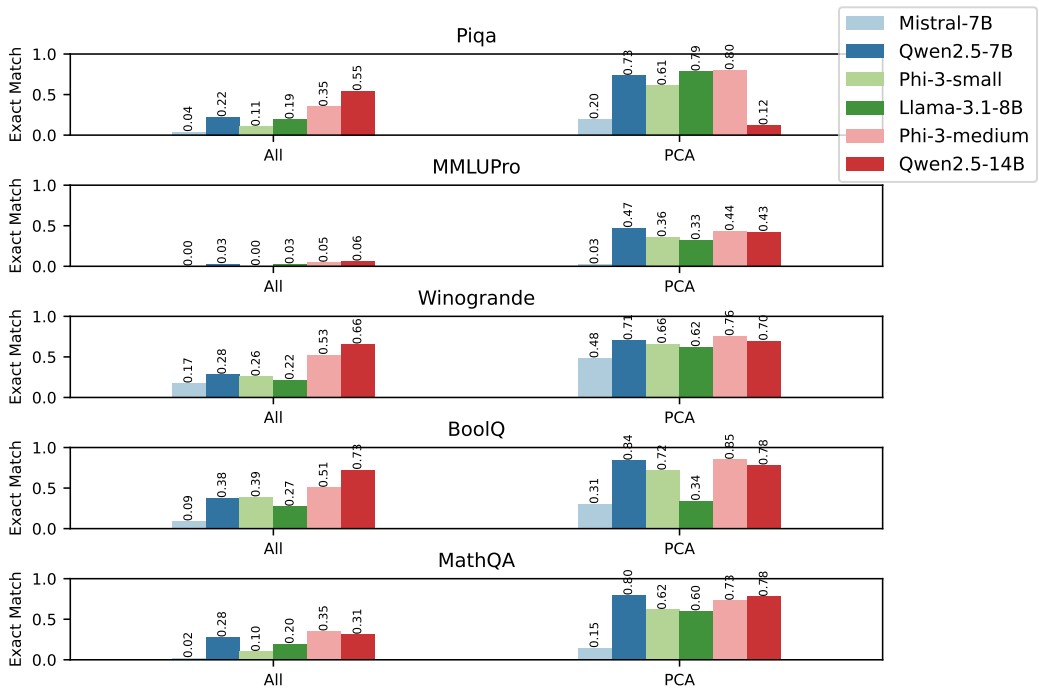

Figure 27: Medium-Scale Models: Performance variation of exact match scores for the Operations on List (Conditional) instruction category compared to its corresponding performance on *print_correct_answer* (PCA).

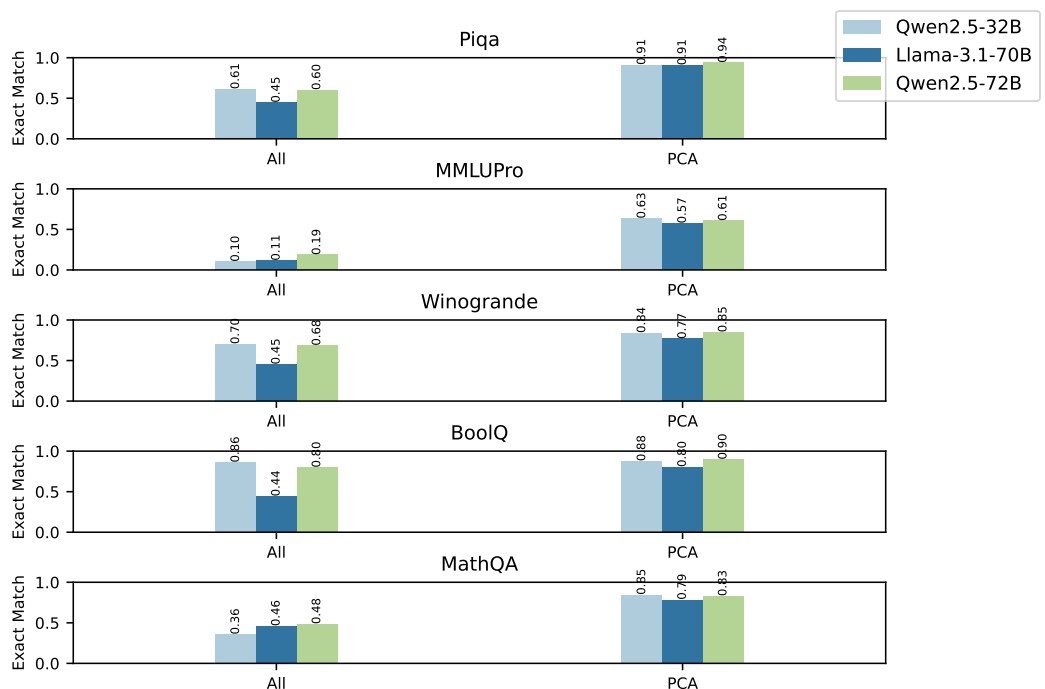

Figure 28: Large-Scale Models: Performance variation of exact match scores for the Operations on List (Conditional) instruction category compared to its corresponding performance on *print_correct_answer* (PCA).

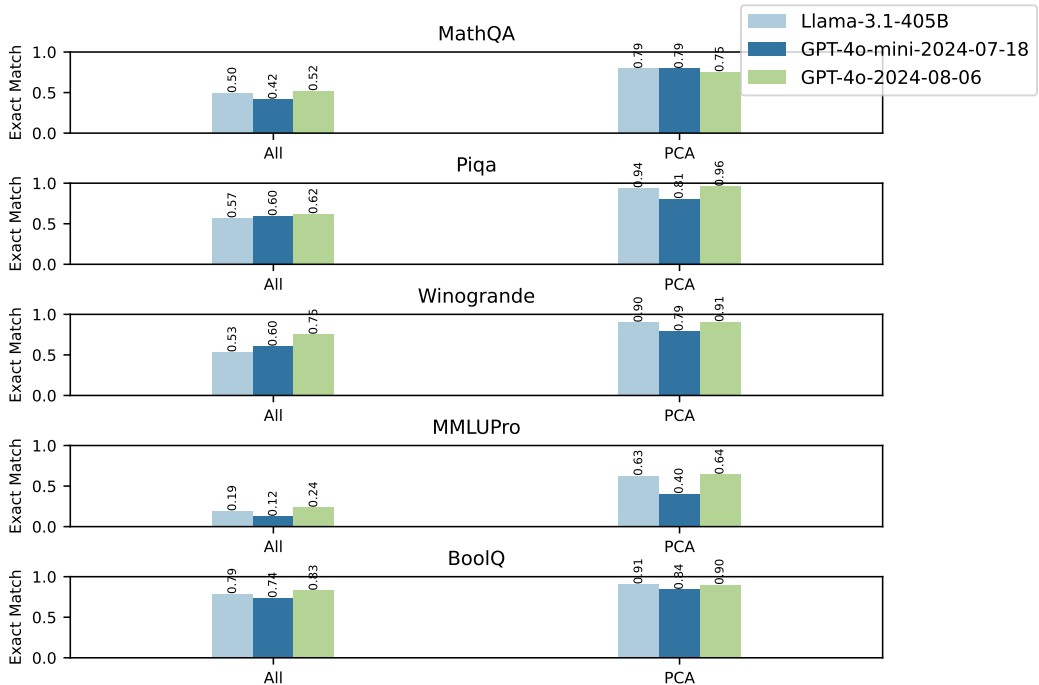

Figure 29: Frontier Models: Performance variation of exact match scores for the Operations on List (Conditional) instruction category compared to its corresponding performance on *print_correct_answer* (PCA).

## D.4 Instruction Specific Results

In this section, we report results at the individual instruction level across knowledge tasks. Figures 30 and 31 show the average performance of models on each instruction in our Full and Lite Benchmarks respectively.

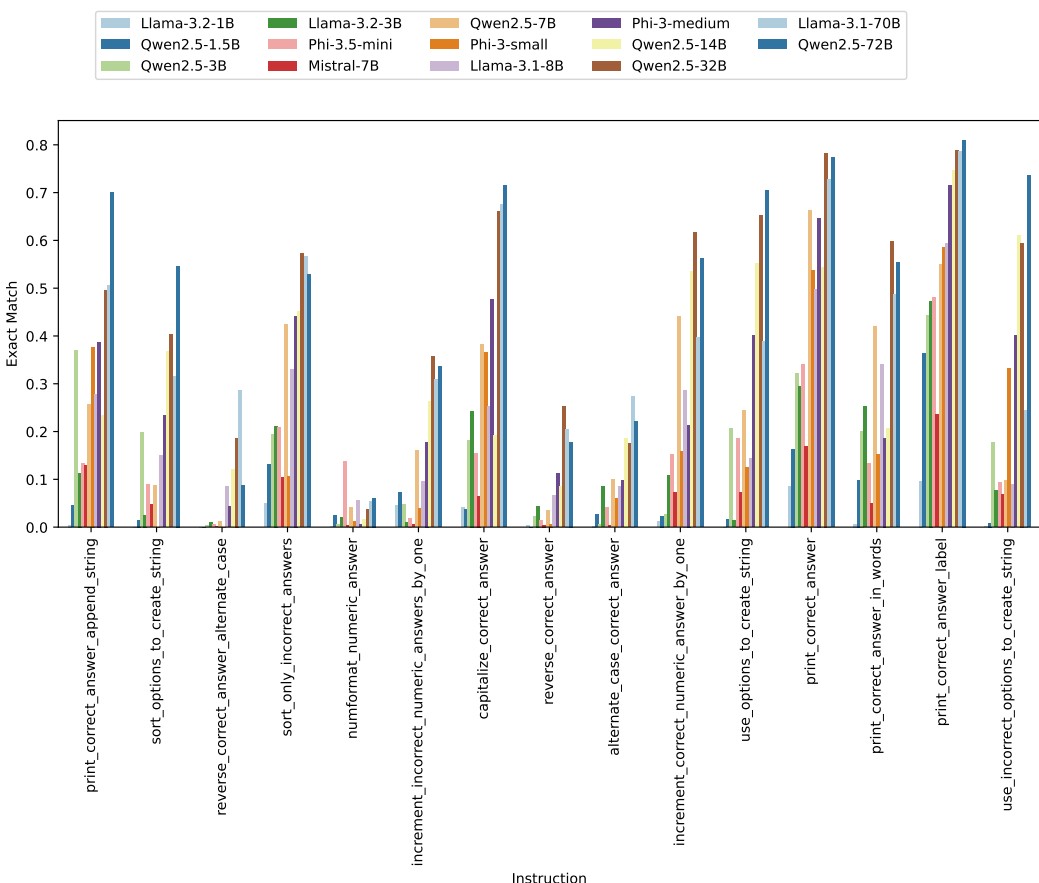

Figure 30: Performance variation of exact match scores on individual instructions across models on Full Benchmark

## D.5 Error classification

Figures 32, 33, 34 present the classification of errors for the LLama, Qwen and Phi family of models. We also report similar plots for small, medium, large and frontier models in Figures 35, 36, 37 and 38 respectively.

### D.5.1 Influence of Parameter Size

We report the performance on Full Benchmark for models from the Llama family and Qwen family of models (Figures 39 and 40). We observe a consistent pattern of improvements in instruction following-ability with increase in model capacity for the Llama family. However, this is not the case for Qwen family of models. Specifically, for some instructions like *print_correct_answer*, *print_correct_answer_label*, *sort_only_incorrect_answers* the Qwen 1.5B model outperforms 3B model. Qwen 3B model is better than Qwen 7B and 14B variants for the *print_correct_answer_append_string* instruction. We consistently see 32B and 72B variants outperforming other models by a significant margin. Performance for the Phi family of models are presented in Figure 41.

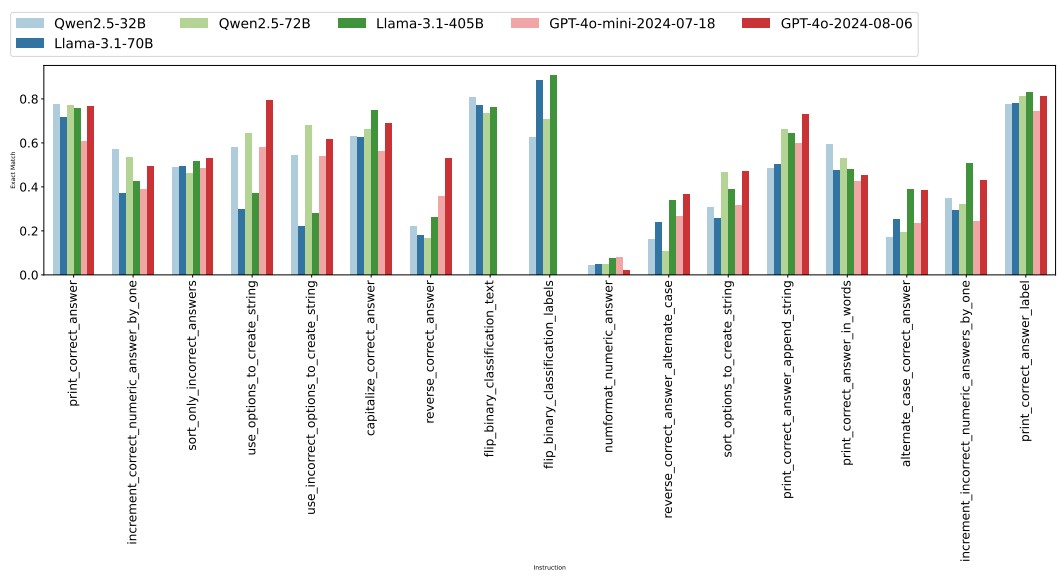

Figure 31: Performance variation of exact match scores on individual instructions across models on Lite Benchmark

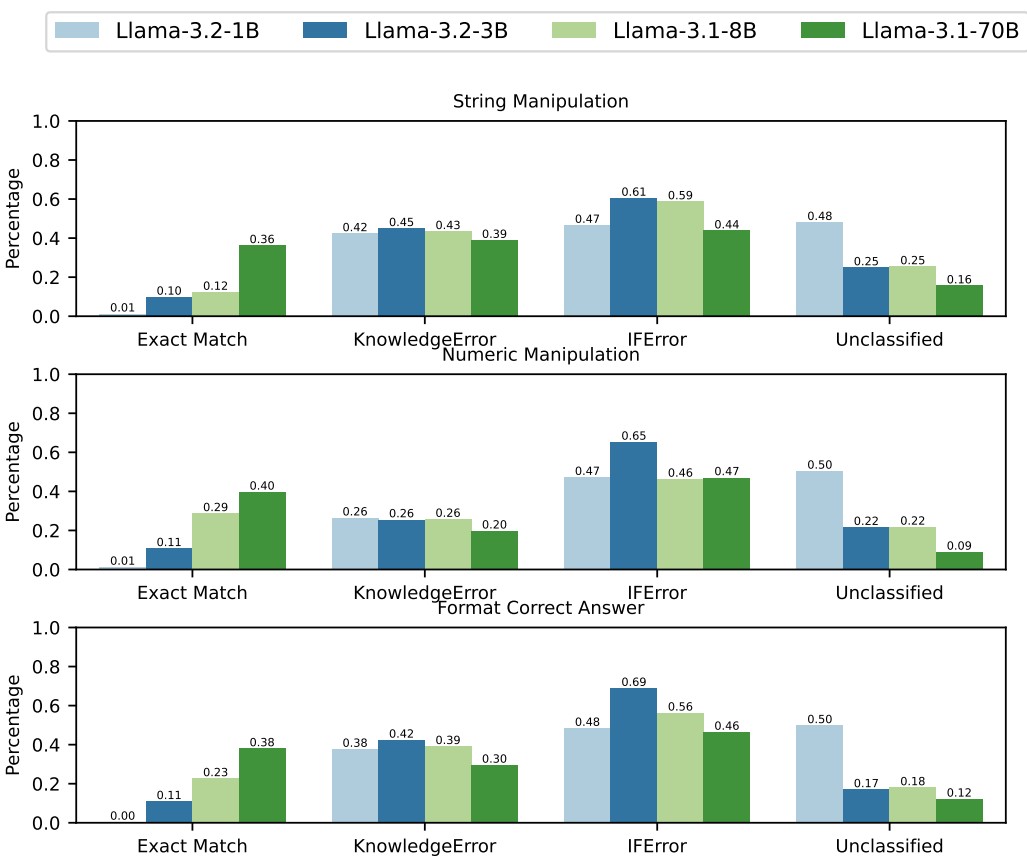

Figure 32: Llama model family: Knowledge Errors and IFErrors

**Presence of Distractors:** We also study the performance of models in the presence of distractors. Figures 42, 43 and 44 report the performance of the LLama, Qwen and Phi family of models in the presence of distractors.

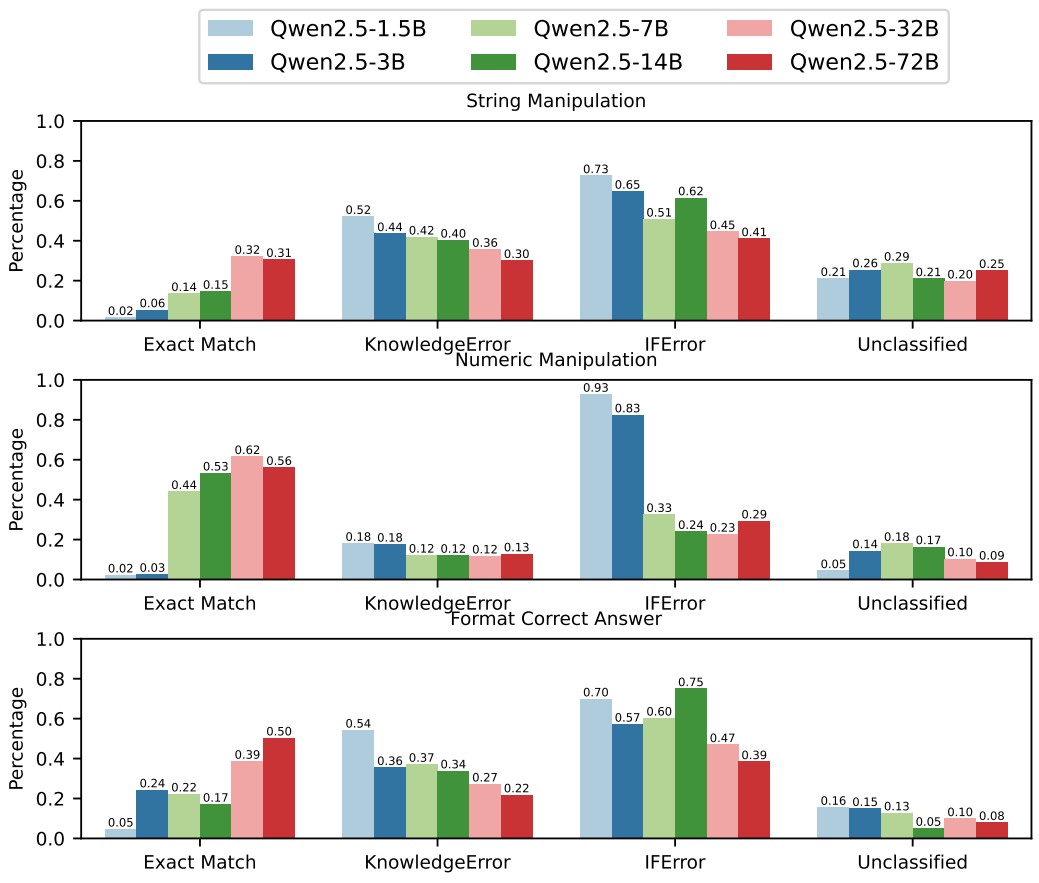

Figure 33: Qwen model family: Knowledge Errors and IFErrors

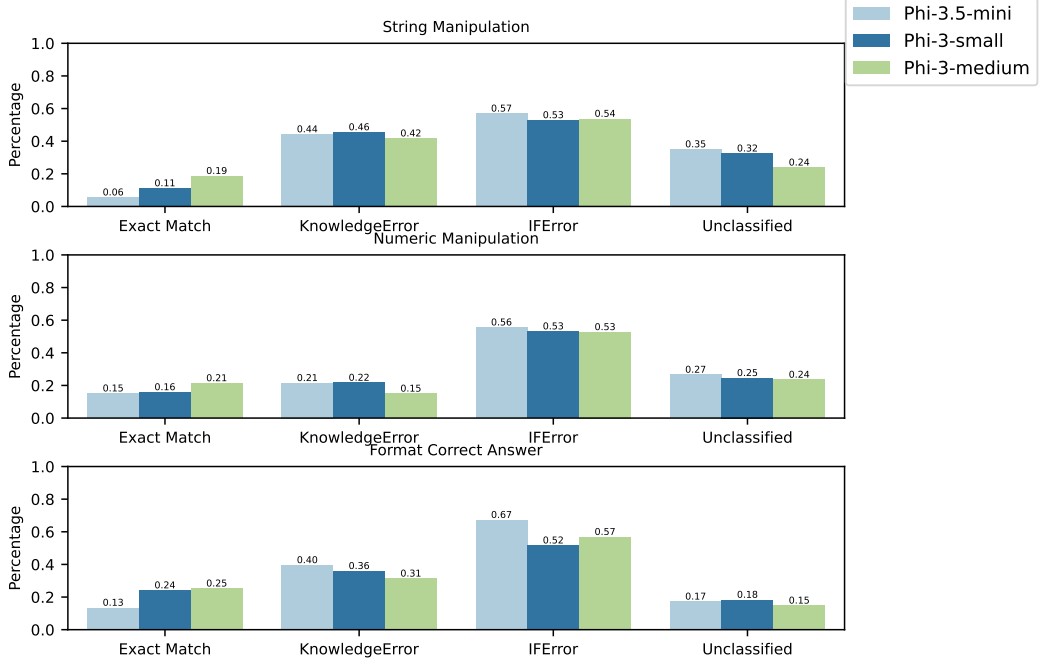

Figure 34: Phi model family: Knowledge Errors and IFErrors

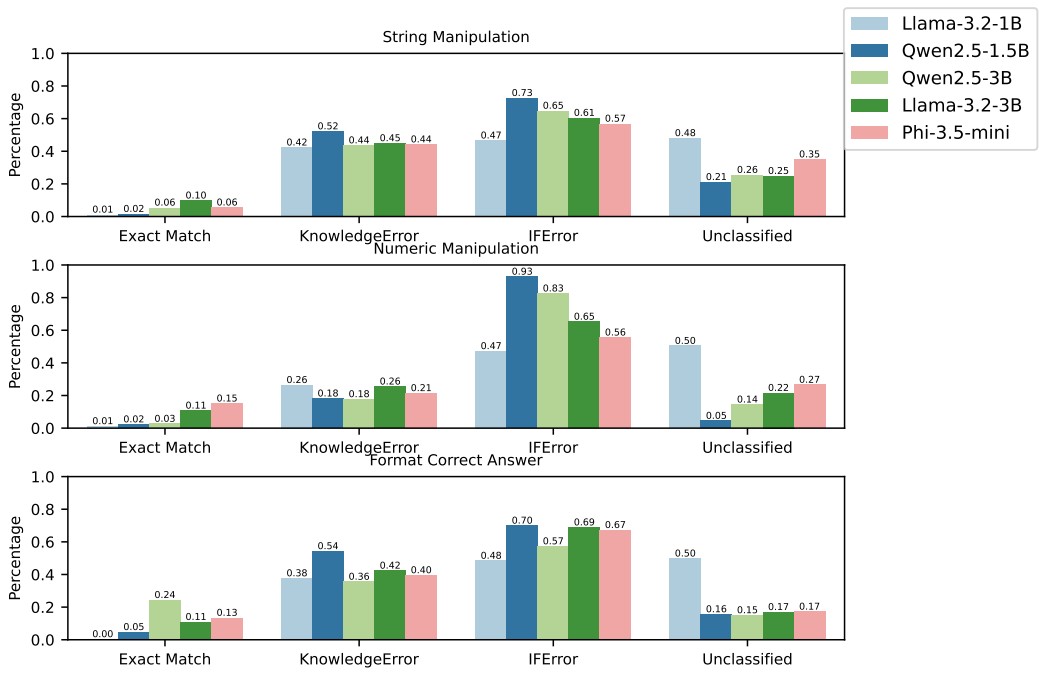

Figure 35: Small Scale Models: Knowledge Errors and IFErrors

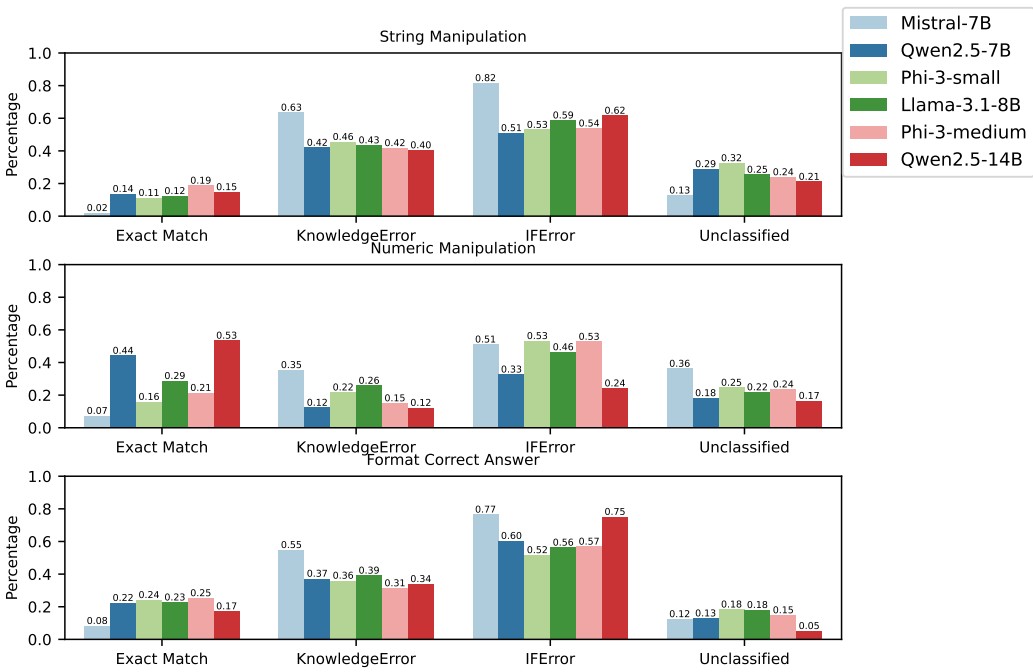

Figure 36: Medium Scale Models: Knowledge Errors and IFErrors

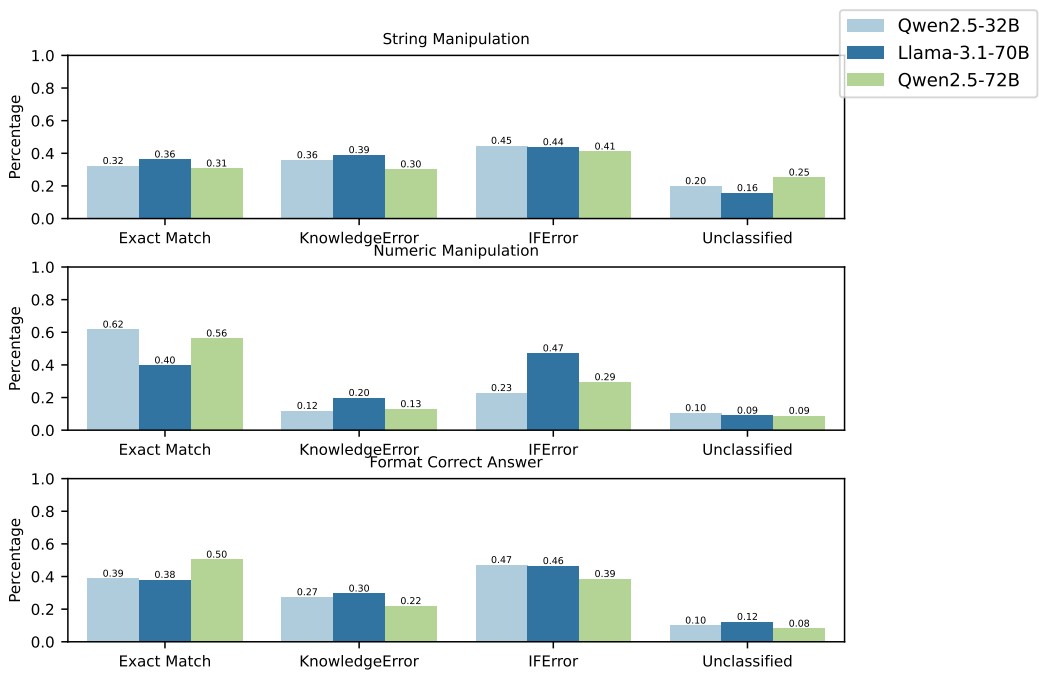

Figure 37: Large Scale Models: Knowledge Errors and IFErrors

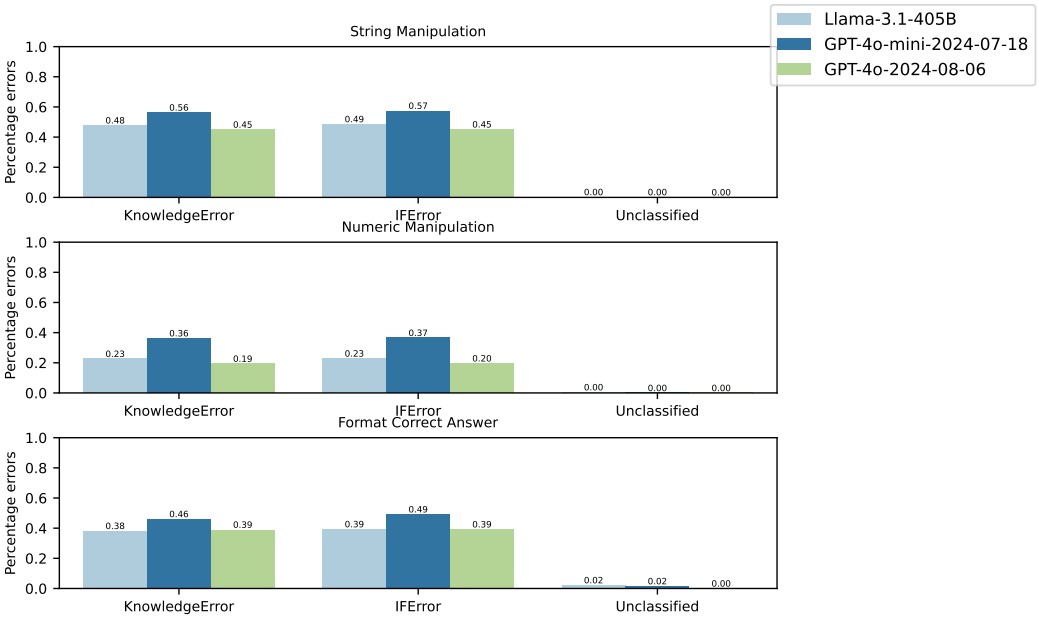

Figure 38: Frontier Models: Knowledge Errors and IFErrors

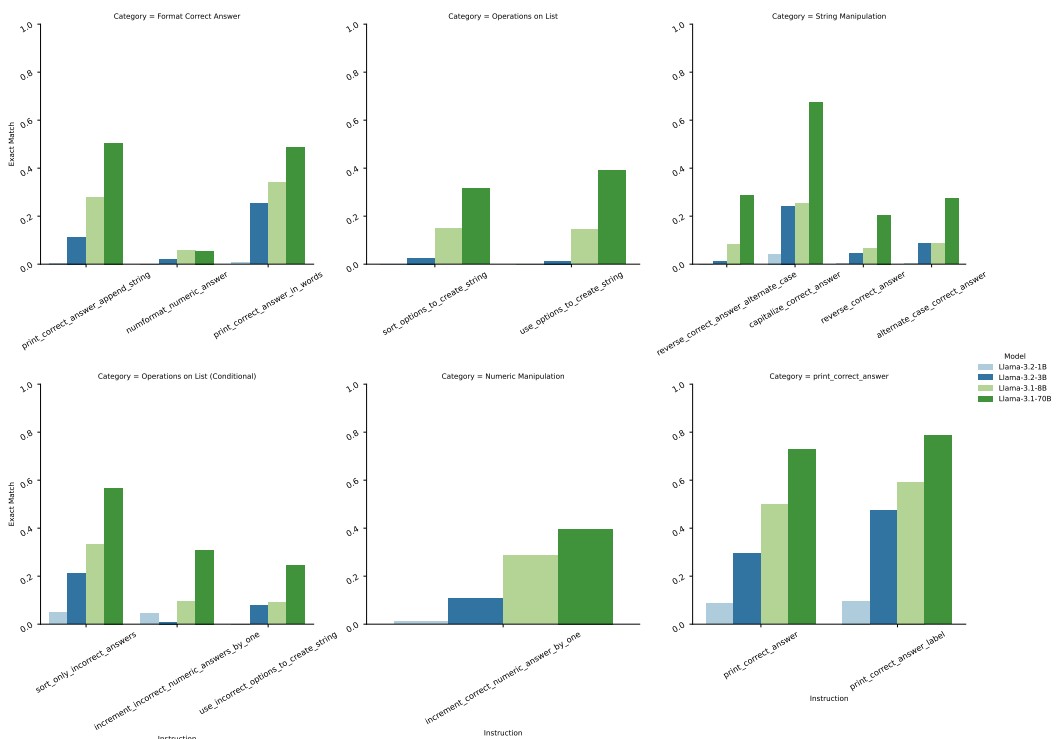

Figure 39: Performance variation of exact match scores for different instruction categories for Llama family of models

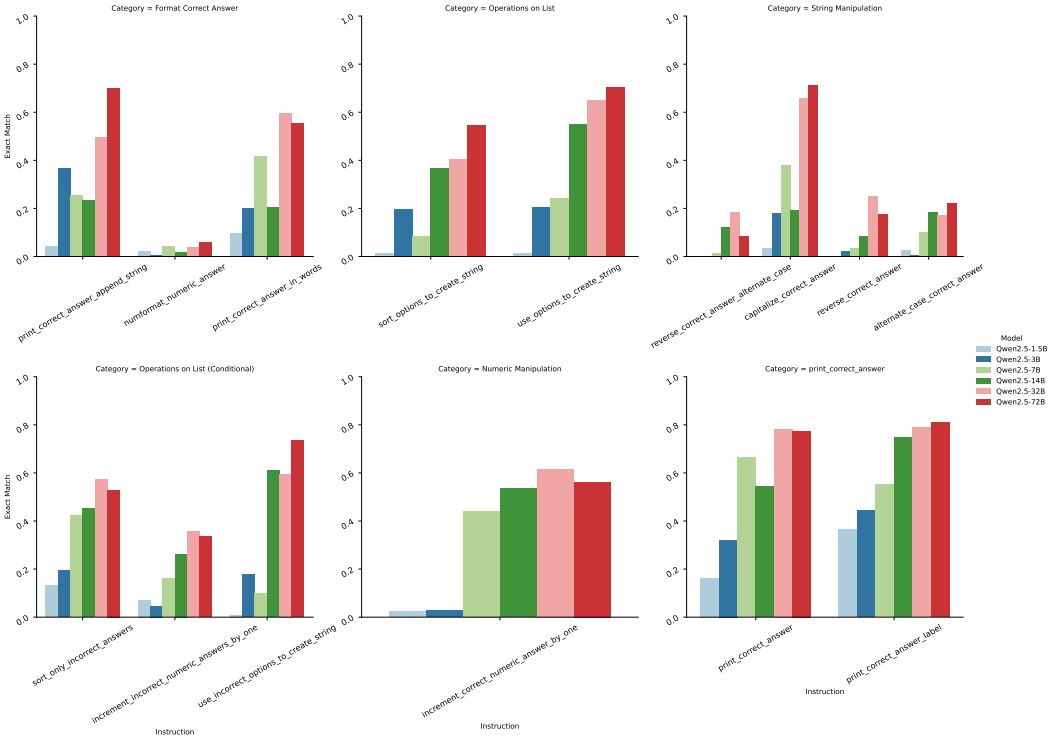

Figure 40: Performance variation of exact match scores for different instruction categories for Qwen family of models

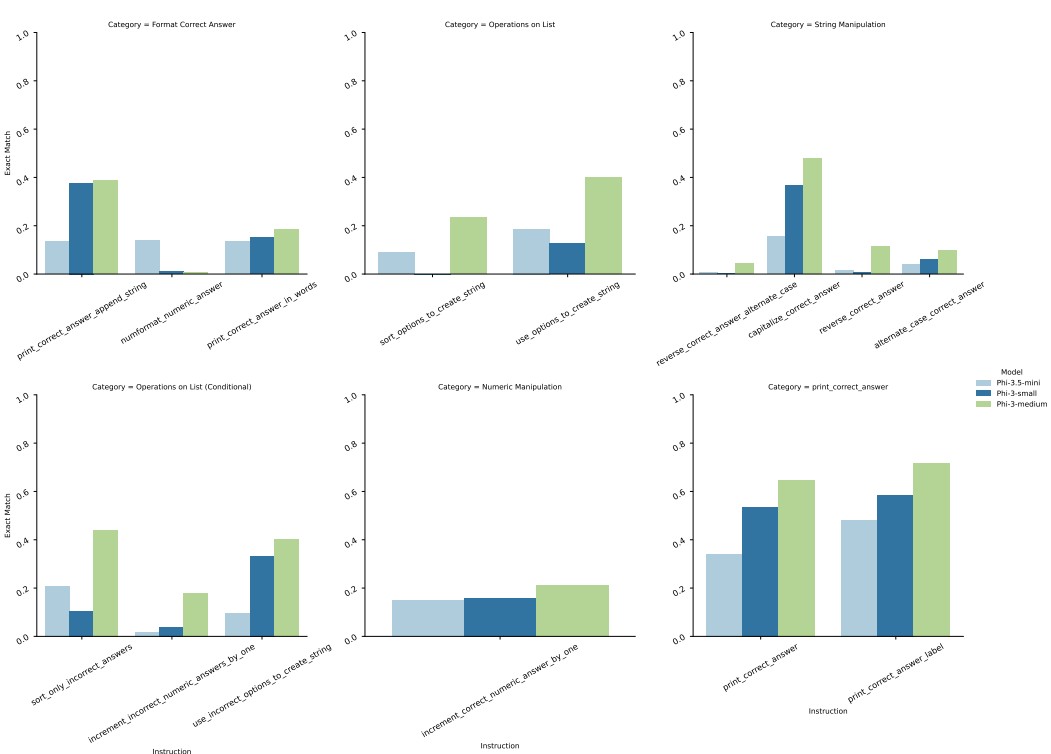

Figure 41: Performance variation of exact match scores for different instruction categories for Phi family of models

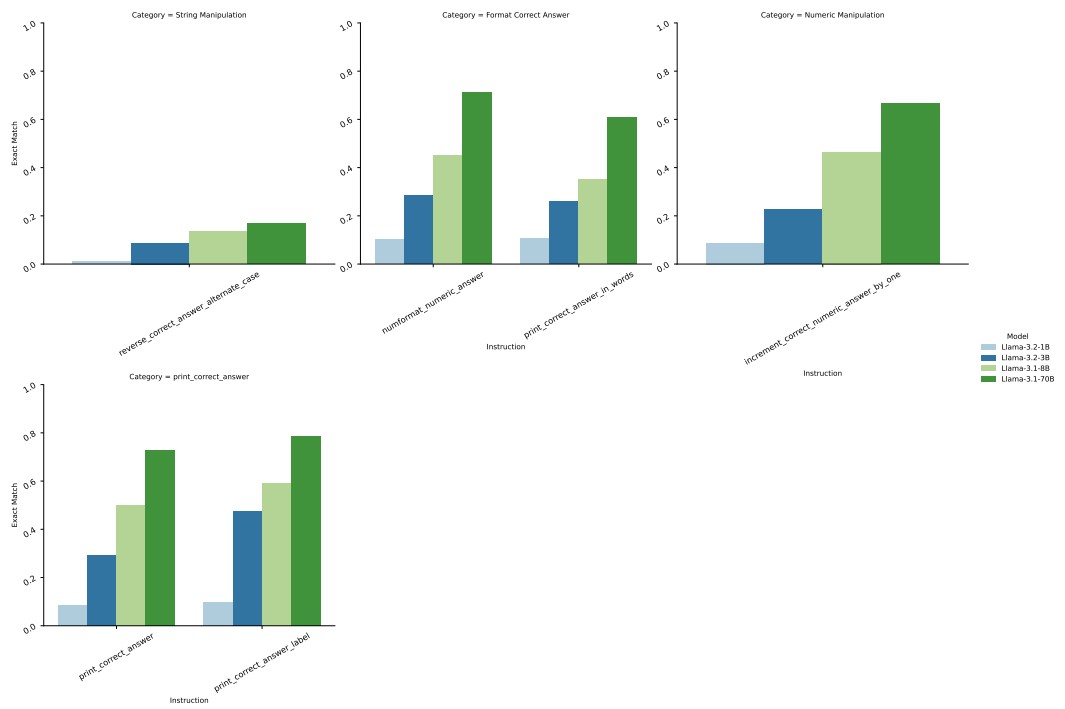

Figure 42: Performance variation of exact match scores for different instruction categories for Llama family of models on the distractor subset

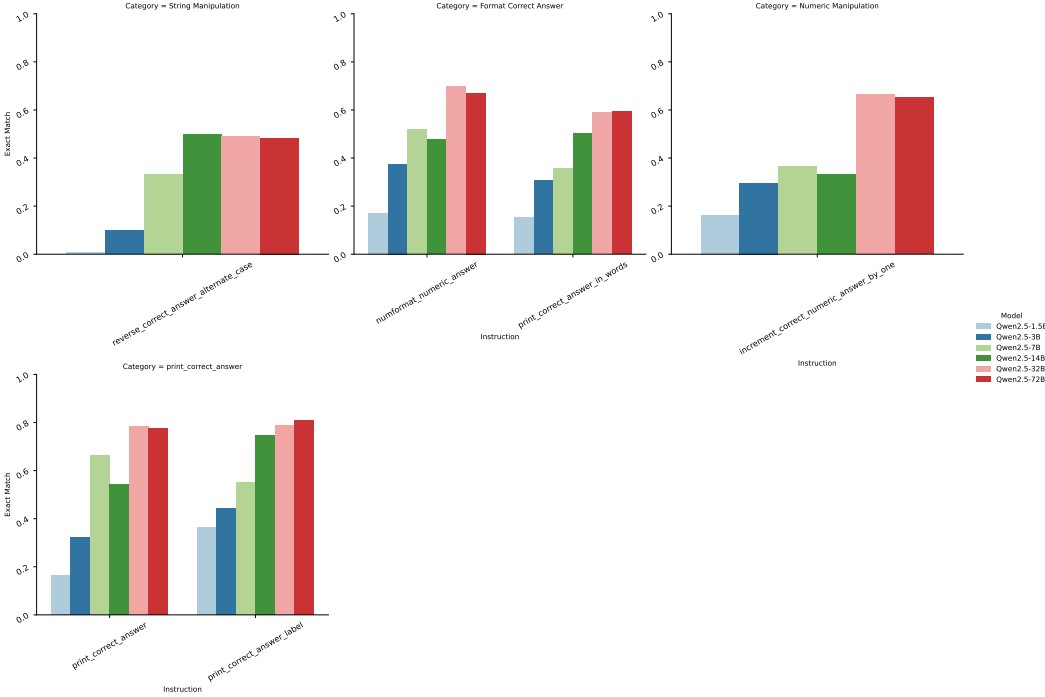

Figure 43: Performance variation of exact match scores for different instruction categories for Qwen family of models on the distractor subset

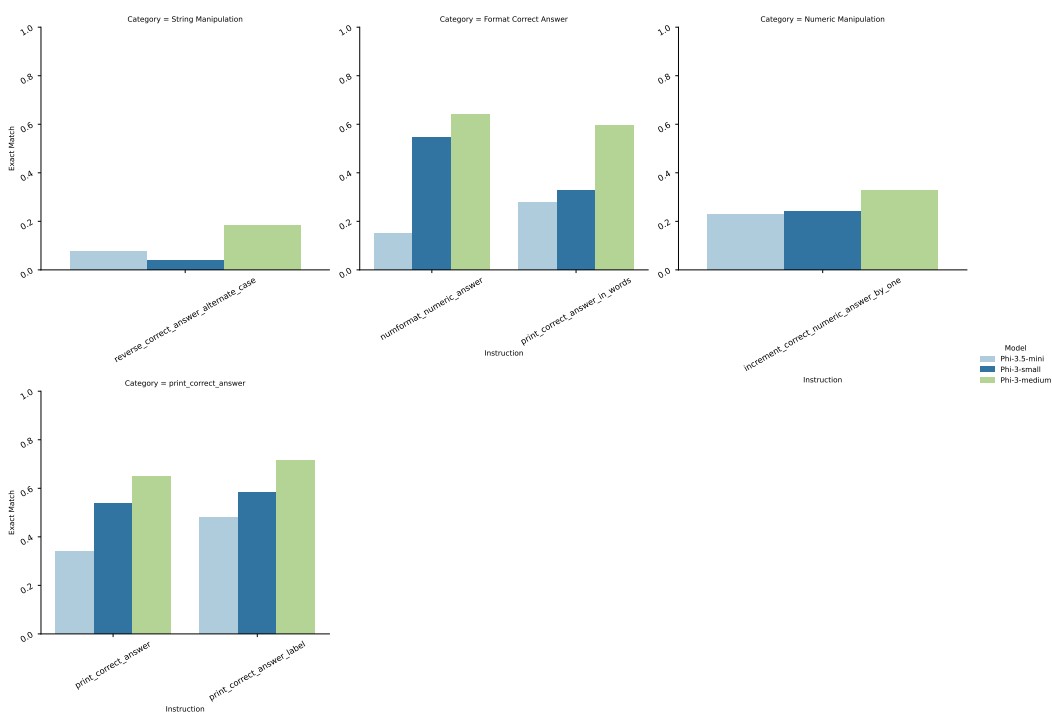

Figure 44: Performance variation of exact match scores for different instruction categories for Phi family of models on the distractor subset

