# OpenReview forum: "KCIF: Knowledge-Conditioned Instruction Following"
_NeurIPS.cc/2025/Datasets_and_Benchmarks_Track — Submitted to NeurIPS 2025 Datasets and Benchmarks Track_

### Official Review · Reviewer_bhP9 · 2025-07-01

**Rating:** 4
**Confidence:** 4

**Summary:**

The paper introduces a benchmark that evaluates the interaction between knowledge and instruction following in Large Language Models (LLMs) by combining simple answer modifying instructions with existing multiple-choice knowledge benchmarks. The benchmark examines the ability of LLMs to follow instructions to manipulate answers in performing knowledge tasks. Results show that all models, regardless of size, experience significant performance drops when asked to follow instructions while answering knowledge questions. This highlights a limitation in current LLMs to combine knowledge and instruction following effectively.

**Additional Feedback:**

A sentence seems incomplete under "Full Benchmark" in Section 3.4: "We select a subset of 1500 samples ... instruction on the same."

**Dataset Code Accessibility:**

Yes

**Dataset Code Comments:**

The data and code is both open-source.

**Ethical Considerations:**

No, there are no or only very minor ethics concerns

**Final Justification:**

The authors clarified that evaluations are performed under CoT, which addressed my main concern about giving the model enough room for explicit reasoning. I have changed my score to the positive side to reflect the value of the proposed benchmark.

**Limitations Weaknesses:**

1. Relationship between "interaction of knowledge and instruction following" and "reasoning" is vague: one could argue that transforming answers after answering a question is a form of multi-step reasoning, and thus is better suited to be evaluated in a setting that permits LLMs to reason step-by-step. For example, it is likely that LLMs trained to perform CoT reasoning (e.g., o-series models and recent Gemini and Claude models) would perform better than non-CoT models on this benchmark. Therefore, to more accurately show the value of this benchmark, the authors should compare the performance in CoT settings versus non-CoT settings, and discuss the relationship between "compound instruction following" and reasoning.

2. Instruction tuning bias: a possible explanation for the performance drop is that the evaluated LLMs are not sufficiently trained on such compound instructions during their instruction tuning phase. The authors could consider evaluating base models (non-instruction-tuned version) with few-shot in-context learning to see if the performance drop is still significant. This would help clarify whether the issue is due to instruction tuning bias or a fundamental limitation in LLMs' ability to follow compound instructions.

**Strengths Contributions:**

1. KCIF is a novel benchmark that effectively evaluates the ability to follow instructions in knowledge tasks. Evaluation results show significant deficiencies in LLMs' ability to follow simple instructions while answering knowledge questions, highlighting an important limitation in current LLMs and the necessity for such a benchmark.

2. The benchmark is easy to implement as it transforms existing multiple-choice knowledge benchmarks by adding simple instructions.

---

> ### Author Rebuttal · Authors · 2025-07-30
>
> We thank the reviewer for their feedback. We address comments from the review below:
>
> **CoT -Reasoning**: We found in our initial experiments, not including chain-of-thought (CoT) in the responses resulted in even poorer performance. Note that all existing evaluations reported on popular leader-boards for source tasks used in KCIF, do indeed use CoT in their official prompts. Thus, in order to keep evaluations comparable with prior work we adapted prompts from popular evaluation frameworks such as Eleuther's eval-harness and prompt the model to use CoT (with output post-processing) in the same way.
>
> **Instruction Tuning Bias**:  We would like to highlight that LLMs are trained on tasks that often involve multi-step reasoning and CoT. We intentionally do not use few-shot demonstrations in our experiments as the instructions are simple, clear and unambiguous and we expect models (especially medium/large frontier) models to follow them. To help models reason, we use encourage models to use CoT via the task prompts.

---

> > ### Comment · Reviewer_bhP9 · 2025-08-06
> >
> > The authors clarified that evaluations are performed under CoT, which addressed my concern about giving the model enough room for explicit reasoning. I have changed my score to the positive side to reflect the value of the proposed benchmark.

---

### Official Review · Reviewer_Pfyo · 2025-07-01

**Rating:** 4
**Confidence:** 2

**Summary:**

This paper proposes KCIF, a benchmark designed to evaluate LLMs’ ability to follow instructions in knowledge-based tasks. KCIF highlights the interaction between knowledge and instruction following, which reflects real-world use cases. Based on existing knowledge datasets, KCIF adds 13 types of instructions, including manipulation, formatting, etc. The authors further propose automated error classification, to distinguish knowledge errors from instruction-following errors. KCIF evaluates 19 models. Results reveal that even simple instructions cause significant performance drop, and distractor instructions also hurt the performance. It provides a new perspective to evaluate the instruction following ability of LLMs.

**Dataset Code Accessibility:**

Yes

**Dataset Code Comments:**

The data is available on Kaggle with standard formats

**Ethical Considerations:**

No, there are no or only very minor ethics concerns

**Final Justification:**

In the rebuttal, the authors explained their reasons for not including more tasks, instruction types, and Chain-of-Thought (CoT) experiments. While I still believe that incorporating these experiments would further strengthen the paper, I find the authors' justifications reasonable: (1) open-ended evaluation is costly, (2) adding more instruction types may not yield additional insights, and (3) CoT reasoning is already a widely adopted strategy in LLMs. Although some of my concerns, such as the limited task diversity, instruction types, and absence of ablation studies, remain unaddressed experimentally, the authors' discussion convinced me that the paper’s practical value exceeds the acceptance bar. Therefore, I will keep my score and recommend acceptance.

**Limitations Weaknesses:**

My main concern lies in the construction of the benchmark and the experimental design.
1. **Limited task diversity.** The study focuses only on multi-choice tasks and does not cover other types, which may limit the practicality of the method. It is better for the authors to consider other types of knowledge-based tasks like open-ended questions that requires explanation.
2. **Limited instruction type.** The instructions used in the framework are mostly single-step. To better reflect real-world scenarios, as discussed in the introduction section, it is recommended to include more complex, multi-step instructions (e.g., "Extract the numeric answer, increment it by one, then convert to words").
3. **Ablation on the usage of CoT.** In section 4.1, the authors mention the use CoT reasoning. Since CoT could potentially shift the LLM’s focus, it is better to conduct ablation analyzing the impact of CoT on performance.

**Strengths Contributions:**

1. **New perspective.** This paper explores the interaction between knowledge-based tasks and instruction following, offering a novel viewpoint. The problem addressed in this paper holds significant importance, as performing tasks under specific instructions reflects common real-world scenarios. Therefore, I find the identified need of this paper convincing.
2. **Comprehensive.** The authors add multiple types of instructions to existing knowledge based tasks to construct the evaluation framework, which is a reasonable approach to study the above problem. The proposed framework contains 13 instruction types across 5 datasets. Additionally, the authors evaluate 19 models on the proposed benchmark, which is comprehensive. Therefore, I think the big idea and appoarch of this paper are generally sound.
3. **Insights**. This work offers insights into prompt design by revealing the impact of instructions on knowledge tasks.

---

> ### Author Rebuttal · Authors · 2025-07-30
>
> We thank the reviewer for their positive feedback and for acknowledging the novelty and comprehensive experiments!
>
> We address some of limitations/concerns below:
>
> 1. **Task-type Diversity**: We intentionally focused on multiple-choice questions in order to have deterministic, reproducible and verifiable evaluations as opposed to the LLM based judges which are often required for open ended evaluation. LLM judges can have biases on domains, tasks, sometimes offer incorrect reasoning for their scores and are also very expensive to run on benchmarks of this size.
> In contrast, we're able to reliably classify errors that offer insights about failures *without* expensive LLM inference costs. That said, our framework is flexible to support contributions of other task types.
> We hope the reviewer is able to appreciate our intentional choices as a strength and not a weakness.
>
> 2. **Multi-step instructions** : We agree our method can be easily extended to create more complex compositions but given the extent to which models struggled with relatively simple compositions, we chose to study that problem in greater depth. We believe that the failure to perform well on simple compositions reveals weaknesses in modern LLMs and at the same time highlights how model scale is still important (eg: models at different scales may have similar performance when printing the correct answer label but markedly different performance when compositions are applied).
>
> 3. **CoT -Reasoning**: We thank the reviewer for this question -- we found in our initial experiments, not including chain-of-thought  (CoT) in the responses resulted in even poorer performance. Note that all existing evaluations reported on popular leader-boards for  source tasks used in KCIF, do indeed use CoT in their official prompts. Thus, in order to keep evaluations comparable with prior work we adapted prompts from popular evaluation frameworks such as Eluther's eval-harness and use prompt the model to use CoT (with output post-processing) in the same way.

---

> > ### Comment · Reviewer_Pfyo · 2025-08-05
> >
> > The rebuttal addresses some of my concerns, although the authors did not provide additional experiments on task types, instruction types, or ablation studies. Nevertheless, their discussion convinced me that the paper’s practical value exceeds the acceptance bar. I will keep my score that leans towards acceptance.

---

### Official Review · Reviewer_wRtf · 2025-07-01

**Rating:** 4
**Confidence:** 3

**Summary:**

This paper examines the interaction between knowledge/reasoning capabilities and instruction-following in large language models (LLMs), challenging the traditional practice of evaluating these abilities separately. The findings demonstrate that LLMs encounter difficulties when processing simple answer-modifying instructions, such as adjusting text case, modifying numerical values, or sorting options, and are additionally susceptible to interference from irrelevant instructions.
To investigate this phenomenon, this paper adapts existing multiple-choice knowledge benchmarks by incorporating straightforward instructional variations, including text manipulation, numerical adjustments, list operations, and distractor commands. The evaluation covers models across different parameter scales (ranging from 1B to 405B) and reveals a consistent and substantial performance decline across all tested models. Notably, larger models exhibit performance drops of 40-50%, while smaller models experience even more severe degradation, sometimes exceeding 80%.

**Additional Feedback:**

1.The paper contains some improper symbol usage, particularly in the second and third paragraphs of Section 1. Please correct these errors and conduct a thorough proofreading of the entire text.
2.Please enhance the explanation of Figure 1, particularly highlighting which models show significant performance gaps in instruction-following capabilities.
3.The layout settings for Figures 2 and 3 are not visually optimal. Please improve their presentation.
4.In Section 4.1, you mention using the vLLM model for evaluation. What are the specific advantages of this particular model?

**Dataset Code Accessibility:**

Yes

**Dataset Code Comments:**

Accessible links to the implementation code and experimental datasets are provided in the paper.

**Ethical Considerations:**

No, there are no or only very minor ethics concerns

**Final Justification:**

The authors addressed my main concern, I have kept my positive score.

**Limitations Weaknesses:**

In Section 3.4, the authors mention randomly selecting 1,500 samples from each chosen dataset. Could this sample size potentially lead to model overfitting?

**Strengths Contributions:**

This paper makes the following contributions:
(i) Presents an evaluation framework that supports LLM-free scoring to study knowledge-conditioned instruction-following.
(ii) Introduces a novel experimental design enabling automated error analysis (to the extent that LLMs avoid catastrophic generation failures).
(iii) Creates an evaluation benchmark using 5 popular datasets, consisting of 13 instruction types across 5 categories. The benchmark is designed for easy extension to additional tasks and instructions.
(iv) Uses this benchmark to evaluate 19 models (1B–405B parameters) from diverse families, revealing that all models exhibit significant performance drops on simple single-instruction task compositions. Large/frontier models drop by 40–50%, while smaller models degrade severely (sometimes >80%).

---

> ### Author Rebuttal · Authors · 2025-07-30
>
> We thank the reviewer for their feedback and for recognizing our contributions.
>
> We address feedback below:
>
> **Sample size and Over-fitting:** (i) Given the number of tasks and instructions, 1500 samples *per* task is a sizeable number as there is also a multiplicative factor that comes in when one considers instruction categories (Ref: Table 1 in paper). The size of the benchmark is thus large and we wanted to keep inference costs in mind.  We'd also like to add that modern LLMs are trained with hundreds of millions (often billions) of tokens and since our data is meant for evaluation only so over-fitting is not a concern.
>
>
> **Symbols in Intro**: Could the reviewer please clarify which symbols were confusing - we have not used any notation in Section 1 and we believe the reviewer may have meant to refer to a different section.
>
>
> **Explanation of Figure 1**: We discuss Figure 1 in the introduction and Section 4.2.1 - we will incorporate the feedback about highlighting a few models in our final version. Thank you for your feedback.
>
>
> **Figure 2 and 3**: We will include patterns in the bar plots to help improve presentation - thank you for your feedback.
>
>
> **vLLM** : vLLM is not a model, it is an inference engine that helps speed up inference by prefilling and caching attention matrices for repeated computations. We only use this for speedup.

---

### Official Review · Reviewer_s7Bz · 2025-07-22

**Rating:** 4
**Confidence:** 2

**Summary:**

This paper introduces KCIF, a benchmark designed to study the interaction between knowledge/reasoning capabilities and instruction-following abilities in large language models (LLMs). Traditional LLM evaluation benchmarks separate these two capabilities, but KCIF demonstrates that LLMs struggle with simple combinations of knowledge tasks and answer-modifying instructions.

**Dataset Code Accessibility:**

Yes

**Ethical Considerations:**

No, there are no or only very minor ethics concerns

**Strengths Contributions:**

1. Addresses a critical gap by studying the interplay between knowledge/reasoning and instruction-following.
2. Includes diverse instruction types (text, numeric, list operations) and distractor tasks, enabling automated error classification (knowledge vs. instruction-following errors).

---

> ### Author Rebuttal · Authors · 2025-07-30
>
> We thank the reviewer for their positive review and acknowledging the contributions of our paper. We request the reviewer to consider their overall rating for the review.

---

### Note · Authors · 2025-08-12

We thank the reviewers again for their feedback and discussion. We are glad that all reviewers recognized the merits of our work and that we were able to address the concerns raised during the rebuttal.

We also thank the AC, and SACs for an active and engaging discussion.

---

### Decision · Program_Chairs · 2025-09-18

**Decision:**

Reject

**Comment:**

## Scientific Claims and Findings
- While LLMs have been evaluated on reasoning tasks as well as for following instructions, past evaluations have not tested these capabilities in conjunction, as is needed in many practical applications. The KCIF benchmark introduced in this paper is created to address this gap: to evaluate the ability of LLMs to follow instructions while solving knowledge tasks.
- The benchmark is created by augmenting 5 MCQ benchmarks with 13 simple instructions that manipulate the answer (string manipulation, sorting, etc.). Evaluation can be performed programmatically without needing a human or an LLM-as-a-judge.
- The benchmark is used to evaluate 19 models, of scales varying from 1B-450B parameters, and from a diversity of model families.
- The findings are surprising and demonstrate significant performance drops on (algorithmically simple) instructions, whereby even large/ frontier models are shown to demonstrate performance drops of 40%-50%.

## Strengths
- **Importance and Novelty**:
  - The reviewers find that the benchmark addresses a critical gap: interplay between instruction following when coupled with reasoning tasks. The ability of LLMs to obey instructions while solving reasoning tasks addresses a practical relevance, and the KCIF benchmark constructed to evaluate this is novel.
  - The experimental design allowing automated error analysis is found to be novel.
- **Methodological Soundness**:
  - The approach is seen to be reasonable, and the methodology followed sound and convincing.
  - Reviewers liked the comprehensiveness of the benchmark created using 5 popular datasets and 13 instruction types across 5 categories. They appreciated the diversity of instruction types (text, numeric, list operations) and distractor tasks, automated (LLM-free) performance evaluation and error classification (knowledge vs. instruction-following errors).
- **Evaluation**:
  - A diverse representative set of 19 models across a variety of model families and size (1B to 405B Frontier models) have been evaluated.
  - Findings are significant, showing that even frontier models struggle in the benchmark settings. For example, printing the correct answer (PCA) task achieves the best performance of 0.77 by  GPT4-o. Single instruction task compositions lead to performance drops of up to 40%-50% while small models degrade severely, sometimes by over 80%.
- **Reproducibility**: the benchmark is easy to implement as it transforms existing multiple-choice knowledge benchmarks by adding simple
Instructions, and the performance can be programmatically evaluated (bhP9).

## Weaknesses
The following weaknesses were noted by the reviewers:
- **Clarity and presentation**: (wRtf) noted about improper symbol usage and gave several suggestions to help improve the information conveyed through several figures.
- **Formulation**: (bhP9) points out that the relationship between "interaction of knowledge and instruction following" and "reasoning" is vague and not adequately explored.
- **Benchmarks Limitations**:
  - **Limited diversity**: (Pfyo) consider the diversity of knowledge tasks and instruction types handled to be limited and inadequate. The benchmark only handles MCQ tasks and simple, single instructions which may limit its practical utility.
  - **Limited sample size**: (wRtf) queried whether 1.5K samples per dataset will lead to model overfitting.
- **Evaluation Limitations**:
  - (bhP9) suggests that the performance drop may be because the LLMs may not be sufficiently trained on such ‘compound’ instructions during the instruction tuning phase. This should be better evaluated using base models (non-instruction tuned) and few-shot ICL setups.
  - (Pfyo, bhP9) suggested that ablation on the usage of CoT should be performed.

## Summary of Discussions, and Changes
- (s7Bz) Gave a positive score, didn’t identify any weaknesses, and didn’t engage in the discussions.
- (wRtf) Raised concerns about sample size and over-fitting; and some minor issues regarding presentation and clarity. He didn’t engage in discussions but indicated that he’s satisfied with the authors’ rebuttal and retained his positive score. The authors assuaged concerns about the sample size and accepted recommendations to improve clarity.
- (Pfyo) Raised concerns about limited diversity of tasks (to MCQ, for example) and instruction types, and ablations on CoT settings which do remain experimentally unanswered. However, the reviewer finds the justification of such ‘design choices’ from the authors reasonable. He retained his score and recommended acceptance.
- (bhP9) Had raised concerns regarding the vague formulation, ablation of CoT settings, the issue of instruction-tuning bias with a suggestion to evaluate models in the few-shot ICL setup. The authors clarified that the few-shot ICL setup was not used intentionally and clarified that the models were prompted to use CoT. The reviewer was satisfied with the response, increasing the score to recommend acceptance.

## Decision Justification
The paper received 4 reviews which are (eventually) all positive:  4(2) (R.s7Bz), 4(3) (R.wRtf), 4(2) (R.Pfyo), and 4(4) (R.bhP9) with the last reviewer changing his score to the positive side post author rebuttal. The latter three reviewers indicated that the author's responses have (partially) clarified their doubts and they recommend acceptance.

*Overall assessment*: The benchmark does indeed address a critical gap in extant benchmarks: testing of reasoning and instruction following together. The evaluation is comprehensive which shows that in the benchmark settings, all models, including the frontline, have a significant drop in performance. Thus, the benchmark will add value to the field.

However, as a benchmark which may continue to challenge LLMs as they improve, there are significant concerns about the depth to which the benchmark challenges and regarding its longevity (continued utility to the community). These concerns are coupled with additional ones regarding the formulation and the evaluations. Consequently, and after post-reviewing discussions, the AC considers that the research with benefit the research community more by incorporating reviewer feedback and further strengthening the research for a future submission at an appropriate venue.